# Multiphoton Laser Fabrication of Hybrid Photo-Activable Biomaterials

**DOI:** 10.3390/s21175891

**Published:** 2021-09-01

**Authors:** Margaux Bouzin, Amirbahador Zeynali, Mario Marini, Laura Sironi, Riccardo Scodellaro, Laura D’Alfonso, Maddalena Collini, Giuseppe Chirico

**Affiliations:** 1Dipartimento di Fisica, Università degli studi di Milano-Bicocca, 20126 Milano, Italy; margaux.bouzin@unimib.it (M.B.); amirbahador.zeynali@unimib.it (A.Z.); mario.marini@unimib.it (M.M.); laura.sironi@unimib.it (L.S.); r.scodellaro@campus.unimib.it (R.S.); laura.dalfonso@unimib.it (L.D.); 2Institute for Applied Sciences and Intelligent Systems, CNR, 80078 Pozzuoli, Italy

**Keywords:** 3D printing, hydrogels, photo-polymerization, photo-ablation

## Abstract

The possibility to shape stimulus-responsive optical polymers, especially hydrogels, by means of laser 3D printing and ablation is fostering a new concept of “smart” micro-devices that can be used for imaging, thermal stimulation, energy transducing and sensing. The composition of these polymeric blends is an essential parameter to tune their properties as actuators and/or sensing platforms and to determine the elasto-mechanical characteristics of the printed hydrogel. In light of the increasing demand for micro-devices for nanomedicine and personalized medicine, interest is growing in the combination of composite and hybrid photo-responsive materials and digital micro-/nano-manufacturing. Existing works have exploited multiphoton laser photo-polymerization to obtain fine 3D microstructures in hydrogels in an additive manufacturing approach or exploited laser ablation of preformed hydrogels to carve 3D cavities. Less often, the two approaches have been combined and active nanomaterials have been embedded in the microstructures. The aim of this review is to give a short overview of the most recent and prominent results in the field of multiphoton laser direct writing of biocompatible hydrogels that embed active nanomaterials not interfering with the writing process and endowing the biocompatible microstructures with physically or chemically activable features such as photothermal activity, chemical swelling and chemical sensing.

## 1. Introduction

The possibility to reproduce functional portions of an organ in living systems is nowadays one of the most active fields of nanotechnology. In living systems, a large fraction of the organism is devoted to responding to sensation: large parts of our organs have both sensing and acting units. How can we more effectively reproduce these sensing units artificially? Living organisms are predominantly composed of water. As in electronics, we need a good match of impedance: an artificial element that can couple efficiently with a tissue should have a high content of water. This is probably one of the most fundamental reasons for the widespread use of hydrogels in nano-biotechnology and nanomedicine. However, nanomedicine R&D relies implicitly on the assumption that the cell is our minimally responsive unit and therefore the relevant spatial features of tissues are cell-sized (1–10 μm) or slightly below (≅100 nm). Hydrogels should then be spatially structured to this scale, which is within the reach of optical (UV–Vis) imaging. Indeed, one of the most widely used approaches to microstructure hydrogels is photolithography, which allows us to fabricate structures with these fine details. 

Clearly, the above picture is simplistic in many respects, lacks details and needs refinements. Microfabrication of completely synthetic structures has been employed to produce implantable cell niches—for example, in the case of microstructured acrylic niches for stem cells. Methods different from photolithography, such as ink-jet printing [1], controlled mixing and microfluidics [2] and electrospinning, have been applied successfully. Moreover, within photolithography, different approaches exist, with advantages and disadvantages. Single-photon absorption, mainly in the UV range and based on the use of masks, can be easily used to fabricate 2D microstructures. However, with methods such as digital processing lithography, 3D microstructures can also be produced. Nonlinear excitation photolithography is instead inherently a 3D photo-activation process that exploits near-infrared light and produces high-resolution structures at a slower rate than DPL (see Table 1).

Finally, but most importantly for this short review, when we refer to spatial resolution, we do not only address topographic details but also the fineness of the single spatial components capable of a heterogeneous response to chemical and physical stimuli. Moving one step forward, we should consider that in order to recapitulate a functional portion of a living organism, we do not only need to fabricate a 3D platform: we also need to provide it with functional units and to distribute them non-uniformly in space. The concept of 4D printing (3D spatial dimensions and a functional dimension) in the field of nanomedicine has been recently put forward, particularly by the group of De Forest [3]. 

Several works have exploited multiphoton laser photo-polymerization to obtain fine 3D microstructures in hydrogels in an additive manufacturing approach or exploited laser ablation of preformed hydrogels to carve 3D cavities in them. These can be used to guide cells and to act as incubation reservoirs or sensing platforms. Less often, the two approaches, additive and subtractive ones, have been combined. Secondly, not many works have focused on the introduction of active nanomaterials into microstructured hydrogels to endow them with sensing capabilities. This review will focus mainly on these two aspects. In particular, we will explore the recent developments in the photolithographic microfabrication of biocompatible hydrogel-based structures capable of guiding the cell cycle and fate and monitoring the cell state. Recent comprehensive reviews have appeared on the microfabrication of resins (organics, hybrids, renewables, functionalized) by photolithography and light confinement [4], on the laser ablation of hydrogels [5], on the surface patterning of hydrogels [6] and on the use of different fabrication methods for tissue engineering [7]. However, these three aspects, fabrication, ablation and functionalization, can be combined in a more effective R&D strategy for cell manipulation and stimulation, and our aim here is to review the efforts in this direction. 

## 2. Light Interaction with Soft Matter

Light–matter interaction provides us with a variety of methods to probe and/or to manipulate matter. Resonance interactions occur when the energy of one or multiple photons matches an energy gap, and the system absorbs them with a fast (femtoseconds) jump to an excited state. Off-resonance interactions lead to scattering phenomena, which are not of interest here. As a consequence of absorption, we may have re-emission as fluorescence or relaxation to the ground state via vibrations. In the first case, we can obtain information in terms of the time course of point-wise signals or of images of the sample. In the second case, the absorption leads to local heating or chemical reactions. Absorption depends on the applied intensity (power per unit irradiation area), exposure duration, light wavelength and polarization, and material properties. If the intensity is very low, $I\leq \;10^{3}\;W/{\mathrm{cm}}^{2}$, already $10^{4}$ times the sun irradiation on Earth, only fluorescence and heating are observed. Increasing the intensity up to $I≅\;10^{3}-10^{5}\;W/{\mathrm{cm}}^{2}$, melting of the sample surface can be achieved if the absorbing molecule is relaxing mostly via vibrations with the surroundings. It is noteworthy that typical intensities used in confocal microscopy, which are of the order of tens/hundreds of μW per square micrometers or $10^{3}-10^{4}\;W/{\mathrm{cm}}^{2}$, may already cause local heating. At values $I≅\;10^{6}\;W/{\mathrm{cm}}^{2}$, the thermal load induced by light affects deeper layers of the material and it may lead to the formation of vapor. Usually, such conditions are applied in laser drilling or welding of metals. If the intensity is larger than $10^{7}\;W/{\mathrm{cm}}^{2}$, the material can be ablated away in the form of a vapor or gas phase that is itself ionized in the form of plasma. These levels of intensity allow subtraction manufacturing on polymers and hydrogels. 

### 2.1. Photo-Excitation and Photo-Ionization

In order to induce changes in the polymeric matrix, we need to provide it with energy, typically in the form of electro-magnetic energy. Photoexcitation is the process that occurs at low irradiance and it implies the absorption of one or multiple photons (multiphoton absorption—MPA). It promotes transitions of electrons from lower-energy ground states to higher-energy states. The dependence of the absorption cross-section scales with the n-th power of the irradiance, where n is the number of photons simultaneously absorbed: the high nonlinearity of the absorption with the laser irradiance determines the spatial confinement of the absorption and allows the 3D optical manipulation of the polymers. Absorption brings the molecule to an excited singlet state, from which it may convert to triplet states that have larger polarity. In these states, the molecule has a good probability to undergo chemical reactions that may lead to the formation of reactive species. This same phenomenon occurs with higher probability when the molecule in the excited state undergoes further interactions with the laser field, with a transition to higher excited states and possible breakage of a chemical bond. The ionization proceeds via nonlinear processes such as multiphoton ionization (MPI), avalanche ionization (AI), and tunnel ionization (TI) [4]. Ionization may also lead to the formation of radicals or of photo-activated acidic species that trigger a series of cross-linking reactions among monomers. These may be the very same compounds absorbing energy from the light field. Alternatively, photoinitiators (PIs) can be used to initiate photo-polymerization. PIs are compounds sensitive to light (mainly in the UV range) which, upon absorption form reactive components capable of starting a polymerization reaction. Their bond dissociation energy is typically ≅ 3 eV or less [8], contrasting the monomers that require higher energy (larger than 4 eV [9]) to induce a cleavage. As an example, amine groups, very frequent in proteins, have a dissociation energy of approximately 390 kJ/mole ≅3.8 eV [10]. Thus, the required intensity I to excite a PI molecule is lower than that of the monomers to be cross-linked. Nonetheless, at least for commercial photoinitiators, the two-photon cross-section $\sigma_{2}$ is not high, rarely above 10 GM (1 GM = $10^{-50}\frac{cm^{4}s}{molecule\;photon})$. Let us take two typical cases: either a 0.01 nJ pulses 100 fs wide at 80 MHz repetition rate or 10 nJ pulses 150 fs wide at 1 kHz repetition rate. In both cases, we assume a writing speed of $v_{scan}$ = 100 μm/s (see Table 2). We further assume high focusing of the laser beam on a beam waist $w_{0}≅1\;\mu m$ and a wavelength in the near-infrared range, λ≅800 nm. We can estimate in the first case approximately $8\times 10^{5}$ pulses per dwell time $\approx w_{0}/v_{scan}$, approximately $2\times 10^{-6}$ absorption events per pulse, and only 16 absorption events per molecule. In the second case, we can estimate 10 pulses per dwell time, approximately 1.3 absorption events per pulse, and a number of absorption events per molecule of approximately 13. The two excitation sources have therefore similar photo-cleavage efficiency.

Let us now consider in more detail the process of multiphoton polymerization, which is of interest here because of its intrinsic possibility to lead to a spatial 3D resolution of the deposited energy.

### 2.2. Multiphoton Polymerization

In multiphoton absorption polymerization (MAP), the sample is a prepolymer resin containing a photoinitiator (PI) that can be excited by MAP and should not absorb by single photon excitation in the same spectral range (typically near-infrared, NIR). In the most common resins, polymerization occurs in the regions that are exposed (negative-tone photoresists), though a few positive-tone photoresists based on polymers (in which the unexposed regions are hardened) have been demonstrated [11]. Most of the 3D laser fabrication developments have been done on acrylic and epoxy resins, as reviewed thoroughly recently [12]. We are here reviewing the 3D fabrication developments of smart devices based on hydrogels.

The fundamental component to choose for MAP is the polymer resin, which is composed of the photoinitiator and the monomers. Other components sometimes used are polymerization inhibitors (that stabilize the resin and influence the features’ size), filler polymers (that create hydrogels or increase the viscosity of liquid resins), and, in general, solvents (that assist in casting the resin). Photosensitizers (PS) are added to enhance the energy transfer from light to the photoinitiation moiety and increase the photo-polymerization efficiency. Other additives (simple dye molecules, small peptides, or nanoparticles) are used to introduce new functions to the polymerized structures. 

Most of the synthetic (acrylic or epoxy) polymers can be made quite viscous, or even in the form of amorphous solids, therefore preventing movements during fabrication. Natural polymers, proteins [13,14], or plant extracts [4], instead, do not allow high viscosity to be achieved (the addition of polyvynil pyrrone, PVP, is employed for this purpose [15]) and this makes printing free-standing microstructures difficult. Each of these options has its own advantages and disadvantages. In any case, after exposure, a development step is needed to remove those portions where the resist has not been cross-linked. Development involves washing with one or more solvents, typically alcohols or water, and can also involve additional processing steps such as baking. Sometimes, prebaking is necessary. Not all of these steps are compatible with bio-mimicking compounds such as proteins, present in the hydrogel formulation.

In simple terms, there are two main types of polymer systems that have been advantageously applied in MAP and these can be classified mainly based upon the type of photoinitiator that primes either a radical polymerization or a cationic polymerization. 

#### 2.2.1. Radical Photoinitiators

Radical photoinitiators are divided into Norrish type I and Norrish type II photoinitiators. Norrish Type I photoinitiators are characterized by a homolytic bondage cleavage reaction of the original photoinitiator into two highly reactive radical species initiated by irradiation with light (either UV, at single-photon excitation, or NIR, at two–three-photon excitation). These radicals then initiate the polymerization (Figure 1A). 

Norrish type II photoinitiators abstract hydrogen from a companion molecule, commonly an amine. The donor forms two radicals that, similarly to the type I case, can then initiate the polymerization reaction. In order to achieve 3D structuring in MAP, the requirement is clearly that these photoinitiators are not single-photon absorbers in the NIR range of the laser used for structuring.

Three major issues are limiting the widespread use of MAP in biotechnology: the cost of the femtosecond lasers and the need to synthesize efficient photoinitiators with complex chemical steps that are often not available to physicists, engineers, or material scientists. An additional issue to overcome for application in biophysics and medicine is the cyto-compatibility of the photoresist, largely determined by the photoinitiator. Indeed, commercial photoinitiators have moderate sensitivity compared to most of those developed in the research laboratories, but they still allow for a number of very interesting applications. 

Among Norrish type I molecules, we can list (E)-stilbene, bis(stryl)benzene, naphthalene, biphenyl, and fluorene. Some of these (stilbene, biphenyl, and fluorene) are also biocompatible. The (E)-stilbene and bis(stryl)benzene systems have been thoroughly studied and modified to obtain high cross-sections for two-photon absorption (TPA) [16]. Although these works were initially motivated by research in the field of TPA fluorescence microscopy [16], it was soon shown that a donor–acceptor photoinitiator bearing a cyano-substituted imino core of stilbene [17] allowed high-throughput photo-polymerization ($v_{3D}≅130\;{\mu m}^{3}/s$) of acrylic resins with a femtosecond laser ($f_{R}=$ 82 MHz, $\tau_{P}=85\;\mathrm{fs}$) at low intensity ($I≅80\;\mathrm{kW}/{\mathrm{cm}}^{2})$: the corresponding TPA cross-section was $\sigma_{2}≅500\;\mathrm{GM}$ at 820 nm. Similarly, Cumpston et al. [18] showed that high TPA cross-section molecules, such as 4,4’bis(*N*,*N*-di-n-butylamino)-(E)-stilbene, can be used for MAP with a few hundred microwatts. Films of these resins were exposed to highly focused laser pulses ($w_{0}\;$∼ 0.35 μm, $\tau_{P}=$ 150 fs, $f_{R}=$ 76 MHz) at wavelengths between 730 nm and 800 nm, finding threshold writing powers as low as 200–300 µW and damage powers greater than ∼10 mW.

Naphtyl, biphenyl, and fluorene systems have been used as photoinitiators for MAP of commercial resins [19] without giving rise to hydrogels. However, these compounds, being biocompatible, are in principle usable for the development of biomimetic microsystems. A diphenyl-amino-benzo-thiazolyl-fluorone analogue was used to polymerize acrylic monomers with P ≅1−10 mW, $f_{P}=$ 1 kHz, $\tau_{P}=150\;\mathrm{fs}$: a pulse energy $E_{P}≅$ 1 μJ (an amplified Ti:Sapphire laser was used) was spread over a large spot size ($w_{0}≅$ 10 μm) [20]. A class of TPA molecules bearing tertiary amines as a donor group and a biphenyl or a fluorene for the transmitting electron group were employed for MAP in the visible and in the NIR ranges [21]. They allowed MPL fabrication of acrylic resins with an inexpensive, frequency doubled Nd:YAG laser.

Several commercial photoinitiators for two-photon polymerization are available [18], though they typically exhibit lower efficiency than those reported above and are not directly intended for the fabrication of hydrogel microstructures. For example, resins produced by San Nopco (Japan) [22] or by Japan Synthetic Rubber Co. (JSR, Japan) are often used for MAP. The SCR500 resin (JSR) is composed of urethane acrylate oligomers. Irgacure^®^ 369 and Irgacure^®^ 184 are efficient photoinitiators belonging to the class of benzoyl peroxides. [23] However, even in this case, a regenerative amplified Ti:Sapphire laser ($E_{P}\;≅\;$0.6 μJ) at 800 nm was needed for MAP. Alternatively to commercial ready-to-use resins, sensitive photoinitiators that can be mixed with custom acrylate or methacrylate polymers are Irgacure^®^ 819 (bis(2,4,6-trimethylbenzoyl) phenyl-phosphine oxide), BME (2-methoxy-1,2-diphenylethanone), and ITX (2-/4-isopropylthioxanthone) [20,24]. Fourkas’ group has worked extensively with a commercially available acylphosphine oxide photoinitiator known as Lucirin TPO-L [25,26] that does not have a high TPA cross-section but has a high radical yield.

Several groups have also demonstrated type II radical polymerization by using custom resins. Type II PIs are used with a co-initiator, typically a tertiary amine. In this reaction, the excited PI, forming an exciplex, abstracts a hydrogen atom from the amine, which is followed by electron transfer to the monomer. Campagnola, Pitts, and co-workers have used several xanthene-based chromophores, including Rose Bengal, erythrosin, and eosin, in combination with the co-initiator triethanolamine, for the radical polymerization of acrylates, acrylamides, and biopolymers. [27,28] Approximately 100 mW of a Ti:Sapphire oscillator lasing at 800 nm ($f_{R}=76\;\mathrm{MHz},\;\tau_{P}\;≅100\;\mathrm{fs}$) was required to create microstructures with these PIs.

Other type II dyes are H-NU 470 and coumarin derivatives. H-NU 470 was used with an aryl amine as co-initiator. To polymerize acrylates and methacrylates [24], an amplified Ti:Sapphire laser with $E_{P}≅$ 15 μJ ($w_{0}≅$ 10 μm, P=3 m W) was required. It was found that 7-diethylamino-3-(2′-benzimidazolyl) coumarin, used with a co-initiator of diphenyl-iodonium hexafluorophosphate, is instead much more efficient [29], allowing fabrication at average powers of approximately 500 μW with a Ti:Sapphire oscillator ($f_{R}=76\;\mathrm{MHz},\;\tau_{P}=120\;\mathrm{fs}$) tuned to 800 nm. The threshold pulse energy for polymerization for this resist and this source was $E_{P}≅$ 6 pJ at a scanning rate of 40 μm/s.

Our group has recently used Rose Bengale as a photosensitizer to polymerize bovine serum albumin (BSA) doped with gold nanoparticles [30]. Typical laser and writing parameters were 200–300 mW over a diffraction limited spot size ($w_{0}≅$ 1 μm) of a Ti:Sapphire laser tuned at 780–800 nm. The pulse energy was relatively low, $E_{P}≅\;$2–0.2 nJ, and polymerization was possible even in the presence of plasmonic nanoparticles. However, much care should be taken not to exceed an energy of around 10 nJ in order to prevent bubble formation in the suspension.

A class of synthetic, water-soluble, curcuminoid-derived benzylidene cyclanone dyes have been recently developed by introducing four sodium carboxylate groups into different benzylidene cyclanone moieties and tested for fabrication in organic polymers, proteins, and acrylic resins. The TPA cross-sections are medium–high, 230 GM $\leq \sigma_{2}\leq$ 810 GM (at approximately 820 nm), implying low writing threshold values, of the order of tens of mW (Ti:Sapphire, $f_{R}=80\;\mathrm{MHz},\;\tau_{P}=100\;\mathrm{fs},\;v_{S}≅10\;\mu m/s$) [31]. Interestingly these PIs also allow photodegradation on the same platform, since a damage threshold value of less than 10 mW was found on these proteinaceous photoresists [31].

#### 2.2.2. Cationic Photoinitiators

A cationic PI generates a strong Brønsted acid that reacts with alkenes and heterocyclic compounds (very frequent in proteins). These PIs are catalytic photoacid generators (PAG) that can initiate a chain polymerization. Two wide classes of cationic photoinitiators for MAP have been studied: diaryl iodonium and triaryl sulfonium salts. These dyes generally display a high TPA cross-section. Belfield et al. used these salts alone to polymerize an epoxy resin (Sartomer K-126) [24]. Boiko et al. realized a mixture of diaryl-iodonium hexafluoroantimonate and 2-/4-isopropylthioxanthone for which the threshold polymerization intensities were as low as 1 $\mathrm{GW}/{\mathrm{cm}}^{\;2}$, with a Ti:Sapphire oscillator (P ≅1 mW) [32] and a wide dynamic range (≅100) that hinders photo-fabrication and photodegradation on the same platform. Other groups have reported custom-made PAGs. Ober and co-workers [33] used the previously characterized coumarin iodonium salt [29] and tested it with PDMS, though they did not provide any irradiation parameters. Marder, Perry, and co-workers modified a series of high TPA cross-section molecules based on a bis(styryl)benzene core with sulfonium ions to obtain PAGs [34] for positive-tone photoresists based on epoxy resins. In summary, PAG initiators have not been specifically applied to the fabrication of hydrogels, though they are interesting for the possibility to offer photo-polymerization and photodegradation on the same platform.

#### 2.2.3. Materials for MAP

Hydrogels represent a fundamentally different type of resins than the those most widely used for MAP fabrication: due to the high content of water, they can undergo easily dehydration during long fabrication procedures and have low viscosity. For these reasons, the fabrication protocols in hydrogels must follow a “continuous scanning” approach and overhang structures cannot be easily produced. Though this review concerns 3D fabrication of microstructures in hydrogels, the best results in terms of nanodevices and sensors are obtained by mixed use of hydrogels and hybrid organic polymers. Mixed fabrication protocols, employing inorganic resins to embed softer hydrogels, have also been reported [35,36].

These hybrid composites have many advantages: they polymerize very quickly compared to other resins due to the high TPA efficiency, form tight connections, are able to resist swelling during the post-fabrication development, undergo limited shrinking upon drying, and can sustain overhang structures. However, ethanol, which is the most used development solvent for these systems, is not fully compatible with biomolecules: mixed resists—biopolymer/acrylic—can therefore suffer from protein denaturation during development (if these have not been already been denatured during fabrication).

Organically modified ceramics (ORMOCERs or ceramers) are even more robust and efficient materials for MAP [37,38]. These silicate-based materials, characterized by an inorganic (-Si-O-Si-) backbone functionalized with organic groups such as acrylates or epoxides, have been traditionally used for photonics [37], but they are now attracting interest for biotechnological applications. The organic side chains cross-link the resin into a durable and biocompatible solid material [39,40]. Always based on silicate materials, polydimethylsiloxane (PDMS) [41] is a widely used material to build microfluidic systems and micro-optics [42] that can be polymerized by means of MPA. Ober and co-workers also reported high-resolution (300–600 nm) photo-polymerization of PDMS in a resist in which ITX acted as PI to initiate the cross-linking of dimethyl-vinyl-terminated siloxane components: low laser powers (P = 3–8 mW) were sufficient for the polymerization when large voxel dwell times ($\tau_{V}=\frac{w_{0}}{v_{S}}≅100\;\mathrm{ms}$) were employed [43]. More recently, Panusa et al. [42] reported the fabrication of compact optical waveguides in PDMS through MPA laser direct writing using phenylacetylene as a photosensitizer, without the use of a photoinitiator. This study was followed by the TPL fabrication of a submicron optical waveguide in PDMS using divinylbenzene as a monomer and Irgacure^®^ OXE02 as a PI [44].

Pure hydrogel materials used for MAP span a wide range: we can divide them into polypeptide-/protein-based and acrylate- or vinyl-based materials. In one example of MAP 3D lithography [45], the hydrogel was prepared from a co-monomer solution containing acryloyl-acetone, acrylamide, and *N*,*N*′-methylene bisacrylamide. Methacrylated poly(ε-caprolactone)-based oligomer (PCL) and poly(ethylene glycol) diacrylate (PEGda) were employed for the MPL microfabrication of hydrogels. Koskela et al. [46] accurately characterized the achievable spatial resolution obtained in writing microstructures in this type of resist with a Nd:Yag nanosecond laser ($\tau_{P}=$ 800 ps) as a function of the laser power and the PI (Irgacure^®^127) concentration. The microstructures were further tested for biocompatibility by seeding neurons on them. More recently, Takayama et al. [47] reported the ablation of PEGDA hydrogels, polymerized by UV curing with Irgacure^®^ 2959, then soaked with gold chloride and ablated by means of a Ti:Sapphire laser ($\tau_{P}=$ 100 fs, λ= 800 nm, $f_{R}=$ 1 kHz, objective lens 60X, NA = 0.7). Hyaluronic acid (HA) has also been used to MPL fabricate microstructures, as in the work by Oviasnikov, Liska et al. [48], where a HA photoresist was developed for MPL fabrication, leading to highly versatile, biocompatible microstructures with the possibility to be enzymatically degraded.

A second wide class of hydrogels that can be structured by MAP is based on functional proteins, mainly BSA, lysozyme, collagen, resilin, elastin, and gelatin. Natural structural proteins display useful functions, such as mechanical toughness, pH dependence (swelling), or electron transfer capability. Proteins occur also in multicomponent systems in vivo. Collagen and elastin are often found together in the body so to combine the strength and toughness required for specific tissue functions. By mixing proteins, one can formulate protein-based biomaterials with complementary sets of specific properties and improved processability. Genetic engineering strategies have been exploited to generate hybrids of structural proteins: click chemistry approaches [49] are being thoroughly explored in this field. Many natural proteins have been studied, with distinguishing mechanical, chemical, electrical, electromagnetic, and optical properties. The group of Campagnola has pioneered the MAP fabrication of proteinaceous resists without radical photoinitiators and using Rose Bengal, eosin, or erythrosin as photosensitizers. They first tested these PS on polyacrylamide [27] (λ= 800 nm, $\tau_{P}=$ 100 fs, P= 100 mW, $f_{R}=$ 76 MHz, $v_{3D}=$ 5 $\frac{{\mu m}^{3}}{s}$) and then applied MAP mediated by Rose Bengal to BSA, fibrinogen, and alkaline phosphatase. They exploited two- and three-photon absorption with Rose Bengal and 9-fluorenone-2-carboxylic acid as PS, respectively. Remarkably, alkaline phosphatase retained its enzymatic activity after cross-linking. The degree of cross-linking was measured by the fluorescence loss of labeled dextrans and correlated with the exposure dose [28]. Additional studies of this group in 2004 showed the possibility to microfabricate multi-protein resists [13,50] and to use MAP to fabricate microstructures inside living cells (starfish oocyte) [51], demonstrating the possibility to insert segregation walls into living cells. Rose Bengal co-injected with fluorescent dextran 10kDa acted as a PS with 500 pJ/pulse and 40–100 raster scanning repetitions. These studies paved the path towards a series of more recent works on proteinaceous structures, again based on BSA [30,52,53]. As an example of hybrid fabrication, Su et al. used BSA to fabricate micro-optics on a substrate of PDMS [14]. They used 600 mg/mL BSA, 0.6 mg/mL methylene blue, laser frequency $f_{R}=$ 80 MHz, $\tau_{P}=$ 120 fs, λ= 800 nm, and a value of NA= 1.35, corresponding to an intensity of I = 4–6 MW/$cm^{2}$ and a scanning step of 200 nm (no detail on the scanning speed was given). 

The choice of materials for cationic polymerization is more restricted than for the radical ones. They are mainly commercial and not particularly suited for fabrication in hydrogels. Epon SU-8 is the most widely used epoxy polymerized by MAP, allowing the production of high-axial-ratio structures and sub-diffraction lithography [54]. SCR-701 (Japanese Rubber Co.) has also been used to produce microgears and nanotweezers [55]. However, none of them produce hydrogels or have been coupled with hydrogels. Generally, SU-8 requires pre- and post-exposure baking, which would not be very useful for hybrid proteinaceous resists, even though it has been also shown that the heating during laser fabrication is as effective as the post-treatment one and results also in finer features than those produced by post-exposure thermal treatments [56]. Interestingly, cationic polymerization PAGs can be used in positive-tone photoresists: this is a promising property to fabricate 3D composite microfluidic devices by mixed additive/subtractive approaches. Marder, Perry, and co-workers demonstrated positive-tone MAP with the initiator BSB-S2 in a random copolymer consisting of tetrahydro-pyranyl methacrylate, methyl-methacrylate, and methacrylic acid units [11,57]. The tetrahydro-pyranyl ester groups are converted into carboxylic acids after the photoacid protolysis, causing them to be soluble in a basic developer. Channels 50 μm×4 μm×4 μm have been made 10 μm below a surface by using an average power of 40 μW from a Ti:Sapphire oscillator (λ=745 nm). Additional details on recent developments in the use of PAGs for cationic fabrication can be found in a recent review [58].

### 2.3. Multiphoton Polymer Ablation/Degradation

Laser ablation and material degradation in the life sciences have been mainly restricted to surgery [59,60,61], but a series of applications to hydrogels have appeared in the last 20 years [5]. Here, we focus on the possibility to create spatially controlled cavities in hydrogels by 3D laser ablation that can enhance the perfusion and exchange of solutes or even guide cell invasion and tissue regeneration. Among the synthetic biocompatible materials that can be degraded by highly focused (laser) light there are poly(ethylene terephthalate) (PET) [62,63], poly(methyl methacrylate) (PMMA) [64,65], polyimide [65], poly(dimethyl sulfoxide) (PDMS) [66], and poly(ethylene glycol) in the form of PEG [67,68] or PEGDA [69,70]. Moreover, natural polymers can be manipulated by laser-induced degradation: collagen [71], agar/agarose [36,72], hyaluronic acid [68], and silk fibroin [73,74]. This technique is suitable for a broad range of applications, from fundamental cell studies to the fabrication of vascularized [69,75], tissue-engineered constructs [76]. 

Laser-induced degradation occurs through the generation of free “seed” electrons. These electrons can undergo two different ionization processes and generate more free electrons: multiphoton ionization (MPI) and tunneling ionization (TI). Exposure to a high-intensity laser imparts enough energy to allow electrons to undergo a transition from the valence band to the conduction band under the laser electric field. For Megahertz repetition rates or above (time between pulses <1 μs), the electrons constantly become excited to a free state via MPI [77]. For a repetition frequency in the kilohertz range, the electrons can relax between pulses. In this case, MPI does not occur but the electric field of the laser, which is larger due to the lower repetition rate, lowers the atomic potential barrier, so that bound electrons tunnel through the barrier and reach a free state (tunneling ionization). In both cases, a threshold intensity is required for ionization to occur and the Keldysh parameter ($K_{y}$) can be used to determine the dominant regime of photoionization [77]: if $K_{y}\;\leq$ 1.5, tunneling ionization dominates; otherwise, multiphoton ionization dominates. The Keldysh parameter can be written as [77]:(1)$$K_{y}≅\;\frac{\omega }{e}\sqrt{\frac{mcn\varepsilon_{0}E_{g}}{P\;}\tau_{p}f_{L}\pi \lambda^{2}}$$
where m and are the electron mass and charge, n is the refraction index, ω and λ are the angular frequency and the wavelength of the light. For an energy gap $E_{g}^{}≅2\;\mathrm{eV}$, λ≅1 μm, $\tau_{p}=100\;\mathrm{fs}$, $f_{L}=80\;\mathrm{MHz}$, we find $K_{y}≅2$, very close to the threshold. The Keldysh parameter depends slightly on the pulse duty cycle. 

In general, the mechanism with which the material is ablated is related to the heat production rate. However, the damage threshold is also affected largely by the thermal energy diffusion in the matrix. For nanosecond and picosecond pulses, the ionization proceeds mainly via TI and avalanche ionization and it implies the formation of luminous plasma, visible at low laser frequencies. For sub-picosecond pulses and high repetition frequencies, high-density plasma is created by the seed electrons and the plasma energy is deposited into the material as a shock-wave, inducing photoablation. For femtosecond pulses (approximately 30–200 fs), multiphoton ionization dominates TI and produces electrons with sufficient energy to cause direct photoablation (Figure 2) [78]. In this case ($\tau_{P}$ ≅ 100 fs, $f_{R}$ ≅ 80 MHz), no visible plasma is formed and the optical breakdown occurs via the sudden deposition of excited electron energy. In polymers and liquids, this leads also to the formation of cavitation bubbles, particularly relevant in surgery [78] but detrimental when photo-cross-linking polymers [30].

## 3. Laser-Fabricated Active Microstructured Hydrogels

Hydrogel microstructures can be post-processed chemically or physically [6] to gain chemical sensing features or to be shaped in suitable forms and become effective platforms for tissue engineering. For these applications, MPL and laser ablation (LA) offer, in addition, the unique possibility to spatially and selectively functionalize preformed 3D hydrogels and to dynamically tune their biophysical or biochemical properties. We will review some of the recent efforts in this direction.

### 3.1. Chemical and Topographic Patterning to Probe and Guide Cells

The extracellular matrix (ECM) is relevant in a number of physiological and pathological processes [79]. The dynamics of cells in ECM-mimicking matrices [80] are mainly determined by specific chemical interactions and by the stiffness of the matrix. One can induce stiffening or softening of the cross-linked hydrogel (post-gelation) or during the cross-linking process [81,82,83], by photo-cross-linking or photodegradation of an already polymerized network. From the physical point of view, the ECM elastic modulus is a parameter that regulates the cell adhesion, spread, and differentiation. Photo-induced stiffening and softening have been employed to test cell reactions in vitro.

There are a few examples of photo-stiffened patterning. Different degrees of photo-cross-linking can be induced by MPA to site-specifically increase the elastic modulus. West and coworkers increased the compressive modulus in PEGDA at specific locations in a PEGDA hydrogel by employing a combination of one-photon mask lithography and MPL [84]. Mehta et al. demonstrated the possibility to pattern PEGDA with sinusoidal strips of a higher-molecular-weight PEGDA (20 kDa) in a low-Mw PEGDA (6kDa): the resulting hydrogel displayed nonlinear and anisotropic behavior that the authors suggested could help in recapitulating better the ECM properties in vitro [85]. They used single-photon lithography, discussing the possibility to implement a MPL fabrication. Other efforts in this direction, made with methods complementary to laser fabrication, including precise chemical design, microfluidics, 3D printing, and non-contact forces, have been recently reviewed by Primo and Mata [86]. Stiffness is always related to the degree of cross-linking that can occur via photo-activation, thermally or chemically. By inserting gold nanorods, Hribar et al. [87] obtained a local increase in the stiffness of a PEGDA matrix: the network was first UV-polymerized and further cross-linked thermally due to the photothermal effect of the gold nanorods induced by highly focused laser scanning. In this way, lines of different degrees of stiffness (17–370 kPa) were patterned (Figure 3). 

In a similar way, Nishiguchi et al. showed recently that a 3D soft actuator composed of thermoresponsive poly(N-isopropylacrylamide) (PNIPAm) and gold nanorods (AuNRs) can be used for 4D printing [88], which is a technology that combines 3D laser printing with dynamic modulation for bioinspired soft materials so that they exhibit complex functionality. Through plasmonic heating by AuNRs, nanocomposite-based soft actuators could undergo programmable fast actuation. Again, by exploiting photothermal nanoparticles, Chandorkar et al. [89] fabricated instead by UV photo-polymerization, thermo-responsive hydrogels that rapidly respond to a light trigger. A soft and patterned thin hydrogel film was prepared by mixing 60/40 mole % of N-isopropyl acrylamide and N-ethylacrylamide (which had a phase transition temperature ≅37 °C) to AuNRs to endow the film with photothermal activity. A fibronectin/collagen I surface coating was implemented to enhance cell attachment. The device exhibited mechanical actuation with frequencies up to 10 Hz when pulsed with an NIR laser and enabled dynamic studies of fibroblast actuation, cell migration changes, and nuclear translocation of growth factors [89].

Local cross-linking, in addition to that induced by focused laser irradiation, can be obtained also by chemical triggers. Stowers et al. developed an alginate system in which temperature-sensitive liposomes encapsulating AuNRs and calcium could be photo-activated by the gold nanorods’ photothermal effect to release $Ca^{2+}$ from the vesicle and cross-link the surrounding alginate [82]. Fabrication of alginate 3D structures has been also recently reported by Valentin et al. [90]. By developing photoresists composed of sodium alginate, photoacid generators, and various combinations of divalent cation salts, a few groups could tune the hydrogel degradation kinetics, the pattern fidelity, and other mechanical properties. With this approach, perfusable microfluidic channels could be fabricated within a few seconds, encapsulating the hydrogel for T-junction and gradient devices. Degradable alginate barriers were also used to direct collective cell migration from different initial geometries [90].

By laser irradiation, one can induce also the spatially controlled softening of hydrogels, though in less direct ways. Hydrogels of PEG–acrylate (PEGA) cross-linked with PEGdiPDA have been widely used with laser stereo-lithography (LSL) and MPL. In these systems, the elastic modulus is locally reduced by surface erosion [83,91]. Kloxin and coworkers studied the spreading of human mesenchymal stem cells (hMSCs) [83] on PEGdiPDA gels, on which they created geometric patterns at reduced stiffness by LSL and MPL. One year later, Tibbitt, Kloxin et al. [67] exploited again a photo-induced chemical modulation of the stiffness. They fabricated a photodegradable hydrogel by redox initiated, free-radical chain polymerization of a photolabile PEG-based cross-linker (PEGdiPDA). The photo-lability was obtained by attaching an acrylated ortho-nitrobenzylether (NBE, Figure 1E) moiety to both sides of a PEG-bis-amine, with a monoacrylated PEG macromer. NBE undergoes two-photon photolysis in the NIR region so that PEGdiPDA cleaves between the NBE and acrylate functionalities, releasing modified PEG and polyacrylate chains with pendant PEG. The authors showed that stiffness modulation could be achieved on and within a gel in the presence of living cells. 

An interesting effect of the matrix stiffness on the spreading of MSCs was further studied by Norris et al. [92], who fabricated micrometer-sized patterns of different elasticity by projection gray-scale photolithography on a mixture of a photodegradable o-Nitrobenzyl derivative of PEG4k-DA (PEG 4000 Mw diacrylate) and non-degradable PEG4k-DA (ammonium persulfate as PI, and TEMED as PS). These are polymerized by redox-initiated radical chain polymerization, and, under exposure to light, photodegradable o-Nitrobenzyl linkages degrade and the PDG−PEG4k-DA chains are released from the network. The authors showed that MSCs aligned orthogonally to the gradient direction and discussed the limitations of projection photolithography with respect to MPL for high-resolution patterning of a similar photoresist [92].

### 3.2. Four-Dimensional Printing and Click Photochemistry

Hydrogels that allow precise control of cells and their 3D microenvironments are needed in tissue engineering [7,93], organoid growth [71,94], and disease modeling [95,96,97]. Besides the possibility offered by direct laser writing to spatially tailor these matrices, for real application in biotechnology or experimental medicine, these structures must be compatible with and responsive to cells, so as to support their 3D growth. Compatibility of microfabrication with cells means that the microstructures allow the cells to grow but also can be fabricated in a cell-laden resist. 

In the past few years [3], click chemistry has been proposed and applied to pattern hydrogels, either with biochemical cues or with physical and biophysical features. This approach exploits elegant, multifunctional, custom-designed constructs, still not widely available: it requires a set of synthesis steps not readily accessible to a wide research audience. The patterning occurs via either photodegradation of preformed hydrogels or photo-activation of chemical reactions between molecules dispersed in preformed hydrogels.

The click chemistry exploits reactions that covalently link two reactants in a straightforward manner at high yield. The copper(I)-catalyzed azide-alkyne cycloaddition (CuAAC) is the prototype reaction of this class. Besides the high reaction efficiency, the true benefit of click chemistry is the low interference of these reactions with many reactive groups (amines, alcohols, and acids) that are common in biotechnology and biophysics. As shown primarily by the De Forest group [3,49], this “reaction orthogonality” allows the independent control of multiple functional groups in a single system and facilitates the 3D fabrication and the post-processing of materials with ever-increasing complexity. In fact, this field is now known as 4D fabrication, the fourth dimension being the possibility to tailor the chemical and physical properties of the hydrogel, after the 3D fabrication, through laser irradiation: for example, cell ligand sites can be introduced selectively in space and time or cell–cell interactions can be triggered at specified locations.

In an early, seminal paper, De Forest and coworkers [3] used a four-arm PEG tetracyclooctyne that reacted with a bis(azide) di-functionalized polypeptide (Azide–RGK(alloc)GRK(PLazide)–NH2) via a copper-free, strain-promoted azide-alkyne cycloaddition (SPAAC) reaction between terminal difluorinated cyclooctyne and azide ($.-N_{3}$) moieties with 1:1 stoichiometry to form, in less than 2 min, a uniform, three-dimensional network (Figure 4A,B). 

The inclusion of a synthetic polypeptide in the gel formulation allows the tailoring of the biofunctionality (for example, enzymatic degradability, integrin-binding ligands, protein affinity binding sites) and the introduction of bio-orthogonal reactive moieties (for example, vinyl groups, azides). Importantly, the timescale and mechanism of the SPAAC reaction permits high viability (95%) during encapsulation of cell lines and avoids the use of copper. Biochemical control in the hydrogel could be obtained in a second step, via thiol–ene photoconjugation (Figure 4C). The lysine(allyloxycarbonyl) amino acid (alloc), which is incorporated into the synthetic peptide, contains a vinyl functionality that is readily photocoupled to thiol-containing compounds, such as cysteine, via the thiol–ene reaction. After gel formation, fluorescently labeled thiol-containing biomolecules were swollen into the network together with a small amount of eosin Y photoinitiator. The reaction was photo-induced and the extent of photocoupling was visualized and quantified using confocal microscopy. Finally, photodegradation is also possible in this hydrogel matrix. In addition to the pendant alloc vinyl functionality, the peptide includes a photodegradable nitrobenzyl ether moiety (Plazide, Figure 4D) that can be photocleaved upon exposure to light (365 nm, single photon, or 740 nm, multiphoton). The authors showed that [3] both the photocoupling and the photo-cleavage reactions were well confined to regions of laser scanning or photolithography: with these approaches, channels of varying depths (150–600 μm) could be created.

More recently, De Forest and coworkers [96] extended this approach to other click chemistry reactions and showed the interesting possibility to covalently decorate cell-laden natural hydrogels with bioactive proteins, including growth factors. These methods were then used to anisotropically govern cell fates in ways not easily obtainable otherwise with high resolution [6]. In 2019, the same group overcame some limitations of the use of azide-based click reactions by developing genetically encoded photocleavable linkers for patterned protein release from biomaterials. Five fluorescent proteins, belonging to three distinct classes (mRuby, sfGFP, and mCerulean), were fused with PhoCl, an engineered monomeric green fluorescent protein that undergoes irreversible polypeptide backbone cleavage upon exposure to visible blue light (λ=400 nm) [98]. The chimeras could be trapped in a gel and be selectively photocleaved, inducing a spatially controlled release of the protein and a loss of fluorescence of PhoCl by mask photolithography [99]. A study on the possibility of MAP-induced photocleavage of PhoCl seems to be missing at present. 

The Anseth group further explored the possibilities offered by click chemistry photoinitiated reactions to template pendant functionalities that allow for the bioconjugation of multiple proteins [49]. They showed that it is possible to spatially and temporarily trigger the exposition of the growth factor to valvular interstitial cells embedded in hydrogels. Three biorthogonal click reactions, a photoinitiated thiol-yne reaction, an azide-alkyne cycloaddition, and a methyltetrazine-transcyclooctene Diels Alder reaction, were used to independently control the presentation of several factors promoting the development of three types of cells: fibroblast growth factor (FGF-2), pro-fibrotic cytokine transforming growth factor-beta (TGF-β1), and bone-morphogenic protein-2 (BMP-2). Exploiting the orthogonal click reactions, FGF-2, TGF-β1, and BMP-2 combinations were patterned into distinct regions on a hydrogel to control the activation of these cells and nodule formation. Again, an MPL patterning study seems to be missing in the literature. 

Other microfabrication approaches, based on photo-clickable chemistry characterized by less demanding synthesis, have been reported on different types of gels. For example, Qin et al. [100] have developed a photosensitive cell-responsive hydrogel composed of cross-linked polyvinyl alcohol (PVA) for multiphoton lithography, which is based on the decoration of PVA with norbornene groups for radical thiol–ene photo-polymerization [97,101] (Figure 1D). PVA–norbornene/DTT (di-thiothreitol) hydrogels could be microstructured under low-intensity UV light (1 min exposure) in the presence of cells. Moreover, these authors succeeded in proving two important dynamic features of the microstructured hydrogels: matrix remodeling of the hydrogel by cells and post-fabrication light-guided decoration with growth factors. Their approach provided fully synthetic matrices that guide cell matrix remodeling, multicellular morphogenesis, and protease-mediated cell invasion. In order to make the gels cell-responsive, DTT was replaced with a dicysteine peptide cross-linker (kMMP=KCGPQG↓IWGQCK, where ↓ indicates the cleavage site), which is known to be susceptible to proteolysis by matrix metalloproteinases (MMPs), and they successfully tested it on MMP-1. Human dermal fibroblasts in permissive gels (containing kMMP) appear highly viable (≈91%) and many cells exhibit a spindle-like morphology, compared to a round morphology when grown on non-permissive (DTT cross-linked) hydrogels. The second approach demonstrated by Qin et al. [100] is based on an initial polymerization of the hydrogels with excess of norbornene groups and by attaching onto the cell-laden PVA matrix different specific cell-instructive extracellular cues in a subsequent multiphoton-induced polymerization. Photo-patterning of CGRGDSP peptides was performed during 3D spheroid culture to obtain well-defined and localized adhesive spots and well-recognizable (Y-shaped) adhesive regions that were indeed quickly colonized. The multiphoton lithography process used by Xiao et al. was very efficient, allowing writing speeds up to 50 mm/s. Therefore, cell invasion could be precisely guided in 3D with micrometer-scale spatial resolution with no harm to the cells.

Kloxin et al. [83] reported the laser-activated decoration of PEG-derived microstructured hydrogels for cell guiding and showed their potential use in the study of cell spreading. The photodegradable moiety was a nitrobenzyl ether derivative largely used in live cell culture and imaging [102,103]. A base photodegradable monomer was synthesized by binding the acrylated photolabile moiety to PEG-bis-amine to create a photocleavable cross-linkable diacrylate macromer (PEGA, Figure 5A,B) from which PEG-based photodegradable hydrogels (Figure 5C) were synthesized. The photodegradable cross-linker was then copolymerized with PEG monoacrylate via redox-initiated free radical polymerization to create photodegradable hydrogels (Figure 5D). Upon irradiation through masks, these hydrogels released modified poly(acrylate) chains and changed their density and stiffness, leading to cell spreading (Figure 5E), while the cell viability was maintained. Cells could also be manipulated directly: a portion of fibrosarcoma cells encapsulated within a hydrogel was subsequently released by degrading channels within the gel using photolithography. Cells were allowed to migrate along the eroded structures (Figure 5E).

Beside the works by the DeForest [3] and Anseth [83] groups, Wylie et al. [104] explored the possibility to pattern spatially in 3D proteins and growth factors able to activate relevant pathways in cell cultures. By means of multiphoton photolysis of coumarin caged thiol groups embedded in an agarose gel, they were able to introduce dynamically in the gel two distinct stem cell differentiation factors, sonic hedgehog (SHH) and ciliary neurotrophic factor (CNTF), by means of orthogonal (non-competing) proteinaceous binding pairs [104]: barnase–barstar and streptavidin–biotin, respectively. In this way, the authors could simultaneously pattern in 3D CNTF and SHH and show that it is possible to control the 3D localization of proteins, to immobilize simultaneously multiple proteins, and to avoid protein inactivation.

### 3.3. Microfabrication for Tissue Engineering 

The application of photo-fabrication and photodegradation is not limited to the development of platforms to study cell migration or to induce differentiation. These tools have been recently applied to the engineering of organoids [105]. Most of these studies have been developed on branched tissues, such as intestinal or mammary epithelial organoids, or on neuronal tissue formation. A few initial studies were performed on collagen gels, preformed by stamping, that allowed the recapitulation of the branching process of mammary epithelial tubules [106] and human small intestinal epithelium, replicating key features of the small intestine in vivo. Wang et al. [107] showed clearly that enteroids growing in a microstructured host matrix, obtained by stamping PDMS structures replicating the villi, could replicate the architecture compartmentalization found in vivo. In these platforms, it was possible to subject the enteroids to gradients of growth factors, obtaining a crypt–villus architecture with cell lineage compartmentalization similar to the one observed in vivo and an open and accessible luminal surface. These studies paved the way towards the use of 3D laser micro-patterning for organoid cultures. A very recent work, presented by the Lutolf group [35], showed the possibility offered by 3D-controlled laser degradation of hydrogels to induce intestinal stem cells to form tube-shaped epithelia. Furthermore, in this case, an accessible lumen could be reproduced together with a spatial arrangement of crypt- and villus-like domains similar to that found in vivo. The microchannels in which the organoid was growing were obtained by photodegradation of a hydrogel mimicking the ECM (75% *v*/*v* of native bovine dermis type-I collagen solution and 25% *v*/*v* Matrigel). The resulting polymerized gel had low stiffness (≅750 ± 50 Pa) and was photodegraded by means of a nanosecond laser system (1 ns pulses, 100 Hz frequency, irradiating in the UV range at 355 nm) equipped with a 10 ×/ 0.25 NA objective. Finer and 3D-controlled structures could be obtained by implementing a multiphoton absorption photoresist.

Regarding the neural cells, the development of real organoids is at a preliminary stage and most studies have demonstrated the possibility to guide the growth and the differentiation of neural cells. It is nowadays clear how deeply surface topographic and chemical features affect neural cell attachment, migration, and maturation [108]. The laser-induced photothermal opening of specific microchannels among microchambers of agarose embedding individual hippocampal cells [36] allowed, for example, the selective growth of neurites at lengths of up to 240 μm (Figure 6A). Most notably, the calcium signals of individual cells growing in connected chambers were synchronized, indicating their active connections (Figure 6B). Doraiswamy et al. [72] went further in the direction of artificial growth of neural tissues by carving 1 cm long microchannels (0.4–0.6 mm wide) in agarose gels. They employed an excimer laser (193 nm, ArF) with a spot size of 30 μm and $E_{P}≅1\;\mu J$, $f_{R}=$10 Hz to obtain smooth, round, and reproducible cuts. By seeding B35 rat neuroblasts in the channels, they could obtain 1 cm long neuron bundles that could be delaminated from the agarose: the thickness of the bundles reproduced that of the original microchannel [72].

The interesting and dynamic in situ guidance of individual neuronal processes has been obtained more recently by Yamamoto et al. [109] by laser modulation of the hydrophilic/hydrophobic character of self-assembled monolayers. These authors started from a simple array of circular dots of hydrophilic/cytophilic (3-trimethoxysilylpropyl)diethylenetriamine; DETA) obtained in a uniform layer of the hydrophobic/cytophobic octadecylsilane (n-octade-cyltrimethoxysilane; ODS) self-assembled monolayer on glass. Then, by erasing with a NIR femtosecond laser (see Table 3) the ODS layer in a hydrogel containing laminin-1, an extracellular matrix protein promoting neuronal adhesion to substrates, they could accurately guide the growth of neurites up to 34 μm in length even along zig-zag pathways. Even though this length is much lower than that obtained by Doraiswamy et al. [72], the method developed in [109] appears to be much more flexible, meeting the need for specific spatial patterns of neurite growth.

### 3.4. Hydrogel-Based Micro-Optics

Micro-optics have a difficult task in overcoming the ancient pinhole camera technique in terms of depth of focus and angular field of view. Pinhole image-forming technology is widely used in the case of bright sources (sun and plasmas) and cameras based on the biomimetic design of insect eyes [110,111]. The diameter of the pinhole for a focal length f should be $d=\sqrt{2f\lambda }$: for f = 150 μm, λ = 0.8 μm; d is approximately 17 μm. Smaller pinholes limit, to a great extent, the amount of collected light, leading to an increase in the exposure time. An improvement in terms of speed with respect to the pinhole camera design is the Fresnel lens, an approach explored by Sun et al. [14,52] in a configuration with transparent even phase plates. Even better in terms of speed are the kinoform lenses, also known as phase-type lenses, in which all the Fresnel zones are transparent but with different phase delays [112]. A further improvement, much more demanding from the point of view of 3D printing, is the compound imaging lens [113], with better light collection power and uniform magnification across the field of view and with compensation of aberrations. This is obviously a great result, though, in this case, the lens is based on hard acrylic resins and hard substrates. However, for medical applications and for smart sensing, it is desirable to base the optics’ micro-fabrication on responsive materials, such as hydrogels. Very few studies have appeared in this field in the literature, as outlined hereafter.

A recent example of active micro-optics can be found in the work by Sun et al. [114], who showed how to fabricate a hybrid poly(N-isopropylacrylamide) (PNIPAm)/cellulose nanofibrils (CNFs) hydrogel that could be used for 3D printing by inverted stereo-lithography. The composition of the resist in terms of PNIPAm and CNFs allows regulation of the lower critical solution temperature (LCST) properties and thus tuning of the optical and bioadhesive properties of the polymerized resist, which becomes transparent below LCST (i.e., approximately 20 °C) and opaque above this value (Figure 7). The PNIPAm hydrogel was cross-linked under the action of 0.5 $J/{\mathrm{cm}}^{2}$ UV radiation (365 nm) and by using Triethylene glycol dimethacrylate (TEGDMA) as a cross-linker, without or with CNF. In addition to the optical properties, it was found that PNIPAm/CNF hydrogels exhibit also switchable bioadhesivity to bacteria in response to CNF distribution in the hydrogels. The PNIPAm/CNF hydrogels possessed highly reversible optical, bioadhesion, and thermal performance, making them suitable for use as durable temperature-sensitive sensors and functional biomedical devices.

The interesting use of protein-based hydrogels for the fabrication of micro-optics that combine optical efficiency, flexibility, and robustness of the optics is shown in the work by Sun, Dong, and coworkers [52]. These authors exploited a BSA-based resist and methylene blue as a photosensitizer to fabricate kinoform lenses (micro-KPL), composed of up to six zones, both on glass slides and on thin PDMS layers, obtaining 50 μm diameter lenses with approximately 350 μm focal length. With N = 2 levels to fabricate each zone, they reached a theoretical optical efficiency of approximately 40%, a figure that can be improved to 81% for N = 4 [112]. The focal length f depends on λ and the first zone radius, $R_{1}^{},$ as $f≅R_{1}^{2}/\lambda$, implying a strong chromatic aberration. The focal length of the fabricated kinoform lenses was approximately 300–440 μm (corresponding to $R_{1}^{}≅$ 12–14 μm). The surface roughness of the lenses, a parameter very important for imaging, was around 10 nm. These lenses represent a valuable balance between biodegradation in natural aqueous environments and sufficient stability in standard biological environments. The optical properties of the protein micro-KPLs were not markedly influenced by the pH value due to their unique opto-mechanical structure. 

Polymeric microlenses with much higher optical quality can be obtained by fabricating plano-convex lenses based on proteinaceous photoresists. Sun et al. [115] reported, in 2012, an accurate study of the effect of the laser writing parameters and of the post-processing on the optical quality and on the stability of the microlenses in aqueous environments. Superior quality was found with respect to the micro-KPLs, at the expense of a longer fabrication time: larger protein concentrations and larger deposited doses (exposure time from 600 μs to 1000 μs, 10 ≤I≤ 24 $\mathrm{mW}/{\mu m}^{2}$) allowed a surface roughness as small as 5 nm to be achieved, with no post-processing. However, for this conventional plano-convex lens shape, a large effect of the pH was found on the lens shape and focal length.

The chromatic aberrations of diffractive laser fabricated microlenses can be reduced by adopting a harmonic diffractive scheme. Tunable harmonic diffractive lenses (HDL) were fabricated in a BSA/methylene blue photoresist by means of two-photon laser fabrication in 2012 [116]. In this case (Figure 8A–C), the difference in the optical paths Δp along adjacent facets of an optical system is an integer multiple of the wavelength, Δp=mλ. In the 2012 study, the authors fabricated HDLs with m = 3, meaning approximately 1.5–2 μm in air. The HDL fabricated up to the third zone had a PSF with a full width at half maximum FWHM ≅9 μm, compared to a FWHM ≅15 μm obtained with a plano-convex lens fabricated in the same photoresist (Figure 8D) [52]. It is noteworthy that the chromatic aberrations of the HDL were much lower than those of the micro-KPLs and the plano-convex lenses [52,115]. It is also important to note that all these studies demonstrate the need for high stability (within 0.5%) of the laser power in order to reduce the micro-optics’ surface roughness.

### 3.5. Microvasculature 3D and 4D Printing

The fields of tissue engineering [7,117] and organoids [35,94,105] have gained much interest in recent years, driven by the rising demand for transplants and implants. Artificial tissues are, however, prone to rejection as foreign objects: one mitigation action is the improvement of the vascularization of the implanted biomaterial. The reconstitution of the microvasculature is therefore as essential step for artificial tissue transplants. Microfabrication is nowadays increasingly used to fabricate complex geometries in hydrogels at the micro-scale [118]. The group of Tien has pioneered the use of collagen to template the growth of the vascular endothelium [119], with a functional response to biological stimuli [120]. However, these first approaches have been unable to produce 3D branched endothelialized networks. Zheng’s group [121] was among the first one to fabricate networks of endothelialized microchannels within matrices of type I collagen. A microfluidic vascular network, engineered by seeding human umbilical vein endothelial cells into a microfluidic circuit formed via soft lithography, spontaneously reproduced in 1–2 weeks.

Many limitations of the microfluidic molding approaches, namely the difficulty to recapitulate accurately a vessel bed, can be overcome by laser-based hydrogel degradation, as performed by Seliktar et al. [122]. Image-guided laser control is used to generate in poly(ethylene glycol) diacrylate (PEGDA) hydrogels complex 3D biomimetic microfluidic networks, and a detailed comparison of a ns to a fs laser system was reported. Slater’s group showed, in a 2016 publication [69], that the vascular bed created by the degradation of the gel and seeding of endothelial cells (EC) can recapitulate the geometrical features and the density of in vivo microvasculature systems and that they can be lined with ECs to generate capillary-like networks. In this work, a 790 nm, 140 fs pulsed Ti:Sapphire laser with a low fluence ($≅37\;\mathrm{nJ}/{\mu m}^{2}$) was raster-scanned on the preformed gel to fabricate the desired geometry by localized degradation (Figure 9A) according to a virtual mask that guided the position of the laser. In this way, it was possible to fabricate two independent, closely intertwining networks that, though never directly in contact, allowed for inter-network transport by diffusion. 

An additional breakthrough is the recent work by the Zheng group [123], in which multiphoton ablation is used to create perfusable micro-vessels at anatomic scale within collagen hydrogels. By seeding them with endothelial cells, the authors created cellularized, organ-specific microvascular structures, such as 3D pulmonary and renal microvascular beds. In this latter case, successful blood perfusion of a kidney-specific microvascular structure was achieved (Figure 9B). In the work by Zheng et al. [123], it is particularly interesting to note the evaluation of the ablation characteristics of three materials among the most commonly used for tissue engineering: collagen (7.5 mg/mL), fibrin (10 mg/mL), and a hydrogel containing 2% (*w*/*v*) each of agarose and gelatin. Channels were laser-ablated with laser powers in the range 107 mW ≤P≤ 244 mW and λ= 800 nm between two reservoirs formed lithographically within collagen or fibrin hydrogels. In collagen and fibrin, laser powers P = 157 mW or above were needed to completely ablate the microchannels.

Similar approaches in which the photodegradation of hydrogels is induced by irradiation with low-energy, ultrashort laser pulses have been applied to mammary tumors and to tissue engineering. In fact, hydrogel scaffolds can mimic well the native extracellular matrix (ECM) environment and their use plays a crucial role in tissue engineering since they can be easily patterned by laser scanning [117]. Friedl et al. [71] employed laser ablation to create small-gauge (around 1–3 μm) channels in the ECM. It is known that the collective invasion of mammary tumor (MMT) breast cancer cells into collagen networks requires matrix metalloproteinase (MMP) activity for collagen breakdown. These authors showed that tracks formed in the supporting gel induced an MMP-independent collective invasion at least for track gauges 3 μm in size. In this case, the microtracks were ablated from fibrillar 3D collagen lattices by means of two-photon excitation at 830 nm with a Ti:Sapphire laser using a 20×, 0.95 NA water-immersion objective (estimated focal spot of around 1 μm) and a focal power of 400 mW, higher than the power levels used by Zheng et al. in similar gels. However, few details on the gel composition were given in [71].

### 3.6. 3D Microfabrication/Ablation in Living Organisms and Cell Cultures

If the first studies and applications of microfabrication with hydrogels have systematically tested the low cytotoxicity and the biocompatibility of the used photoresists and the developed microstructures, nowadays, the state of the art in this field is to directly fabricate microstructures in living organisms, to ablate live tissues, or to encapsulate living cells. The main issue here is that the very same PI molecules are often produce single oxygen and other radicals that are detrimental for the cells.

One of the first developments of 3D microstructured hydrogels for cell manipulation dealt with chemotaxis [124]. Lee et al. used MPL (here due to TPA) to micro-pattern regions rich in cell adhesive ligand (RGDS) within preformed PEG hydrogels embedding clusters of human dermal fibroblasts. Short-duration (20 s) UV photolithography was used to trap living cells in a degradable PEG obtained by inserting a collagenase-sensitive peptide sequence, GGGLGPAGGK, in the form of acrylate-PEG-(GGGLGPAGGK-PEG)n-acrylate (MW ca. 15,700 g/mol). This cell-laden pre-cross-linked hydrogel was incubated in FITC-RGDSK-PEG-acrylate solution and in a PI ((2,2-dimethoxy-2-phenyl acetophenone) and subsequently photo-polymerized in 3D by laser scanning, obtaining the precise location of RGDS binding in the hydrogels (Figure 10A). The RGDSK-PEG-acrylate moiety was polymerized by TPA at 720 nm, 203 $\mathrm{mW}/{\mu m}^{2}$, and 240 $\mu s/{\mu m}^{2}$ scanning rates. When human dermal fibroblasts cultured in fibrin clusters were encapsulated within the micro-patterned collagenase-sensitive hydrogels, the cells underwent guided 3D migration only into the RGDS-patterned regions of the hydrogels.

A second type of approach to cell trapping and sensing/manipulation by means of 3D MPL was given a few years later by Hasselmann et al. [125]. In this study, single fluorescent E. coli cells were trapped by the fast (10–15 s) fabrication of PEGDA hydrogel (with Irgacure^TM^ PIs) pillars (2 μm radius) close to the cell. For TPL fabrication, a pulsed laser source ($\tau_{P}=$ 94 fs, λ= 780 nm, $f_{R}=$ 80 MHz) was used with an average power of 2.64 mW over an estimated laser spot ≅1 μm FWHM. By means of these parameters, fabrication could be performed at a stage speed of 10–30 μm/s. Different cells could be blocked and connected by means of thin hydrogel tubes that could be coupled to laser sources to optically interrogated individual cells (Figure 10B). In a later work, the same group [126] investigated the possibility to trap bacteria by 3D fabrication in a protein-based hydrogel precursor. Laser-induced cross-linking was used to attach microstructures to the hydrogel backbone, resulting in a motility change or complete immobilization of a selected bacterium. In this work, a protein-based composite containing BSA and riboflavin 5′-monophosphate sodium salt hydrate (FMN-Na as PI) was used with 2PP-DLW to attach microstructures to single, preselected bacteria (bacillus subtilis). The very same laser setup used previously allowed, in this case, a lower fabrication power ≅0.9 mW to be achieved at a stage rate of 10–20 $\frac{\mu m}{s}.$

As an example of a study of microstructuring gels loaded with cells, Troymayer et al. [127] employed a cleavable diazosulfonate (DAS) two-photon PI and compared its performance with the commercial PI 3,3′-((((1E,1′E)-(2-oxocyclopentane-1,3-diylidene) bis(methanylylidene)) bis(4,1-phenylene)) bis(methylazanediyl))dipropanoate (2PCK) [128]. Since the efficiency of this PI is not very high, they had to use larger PI concentrations, reaching, however, good cyto-compatibility, as demonstrated by the encapsulation of adipose-derived stem cells (ASC) in gels photo-polymerized by MAP of the diazosulfonate PI. As claimed by the authors, this is probably due to the reduced singlet oxygen formation of DAS compared to 2PCK. The writing threshold power for this PI was approximately 60–80 mW (equivalent to I = 420–560 ${\mathrm{GW}\;\mathrm{cm}}^{-2}$ for the microscope objective used) at scanning speeds ≅100 mm/s, employing a Ti:Sapphire laser tuned at 800 nm ($f_{R}=$ 80 MHz, $\tau_{P}=$ 70 fs at the sample plane). It was remarkable that cells could replicate up to 140% of the initial seeding number after 5 days for gels polymerized with power levels lower than 100 mW.

Very recently, a thorough analysis of the possibility offered by a recombinant collagen type I protein (RCPhC1), endowed also with RGD sequences for cell adhesion, has been presented by the Wien group [129]. In this case, the protein was functionalized with photo-cross-linkable methacrylamide (RCPhC1-MA), norbornene (RCPhC1-NB), or thiol (RCPhC1-SH). The 3D MPL was performed by raster scanning at high speed with a Ti:Sapphire laser delivering 70 fs pulses at repetition rate = 80 MHz, tuned at 720 nm. The layer and line spacing were quite loose, 0.5 μm, and the writing speed was high, up to 1000 mm/s, allowing photo-polymerization in proteinaceous photoresists in the presence of living cells. This implied the use of a highly efficient photoresists and PI, which, in this case, was tetrapotassium 4,4′-(1,2-ethenediyl)bis[2-(3-sulfophenyl)diazenesulfonate] (DAS). Altogether, the authors could use relatively low average laser powers of 10–100 mW. The main result in terms of the viability of cells encapsulated in laser-printed hydrogels was that nonbornene’s step-growth was much better than the chain-growth of methacrylate and thiol polymerization. Moreover, cell division could be observed for immortalized human adipose tissue-derived stem cells (ASC) labeled with GFP (Figure 10C) and was dependent on the gel cross-linking as modulated by the percentage of the photo-cross-linkable functionalities.

Three-dimensional MPL fabrication has been applied also for trapping and manipulating larger organisms. A first approach to MPL microfabrication in the presence of a full living organism was presented by the Stampfl group at the TU Wien in 2012 [132]. These authors used *Caenorhabidtis elegans (C. elegans)* as an inexpensive and ethically acceptable biosensor for animal organisms in predicting the acute lethality of the laser processing instead of performing time-consuming cytotoxicity tests. The fabrication was performed by means of a Ti:Sapphire laser ($\tau_{P}≅\;$100 fs, λ= 810 nm, $f_{R}=$ 73 MHz, P = 400 mW) and of a water-soluble PI. Remarkably, the raster scanning polymerization of a woodpile structure was performed around the living animal and through the animal cuticle itself, resulting in the structure being attached to the exterior of the worm with no apparent harm to it or any visible reaction.

Three-dimensional MPL fabrication has even been performed in vivo. In a recent paper, Urciolo et al. [130] devised PEG-derived photo-cross-linkable polymers with sufficient efficiency and biocompatibility to allow photo-polymerization in vivo. The cross-linkable moiety, a specific coumarin dye (7-hydroxycoumarin-3-carboxylic acid, HCC) selected among 58 derivatives, was introduced in four- and eight-arm model PEG compounds in order to reach an efficiency such to allow 3D MPL fabrication at average powers of 0.8 mW in the range of 700–850 nm (Ti:Sapphire laser; scan velocity was 0.8 mm/ms, pixel dwell time 2 μs, and z spacing 1μm). A volume of 1 ${\mathrm{mm}}^{3}$ could be polymerized in around 30 min. The HCC–polymer solution was injected locally into mice and polymerized by raster scanning the focalized pulsed near-infrared laser. In this way, it was possible to efficiently fabricate isolated 3D objects inside the dermis of the mice (Figure 10D). The authors were also able to photo-polymerize in intact skeletal muscle or in mouse brain: in these cases, the HCC–PEG polymer solutions were injected below the epimysium or meninges, respectively. For example, 3D hydrogels were fabricated at the surface of muscle fibers without evident alteration of the overall morphology of the muscle fiber. It should be noted, however, that the authors detected the recruitment of macrophages to the site of hydrogel cross-linking, even though there was no evidence of histological changes.

### 3.7. Biomimetic Responsive Microstructures

Hydrogels have an intrinsic pH dependence that can be and has been exploited to develop actuators. In addition, there are a few polymers whose chemical structure depends on temperature or light and which, combined with biocompatible polymers, constitute good photoresists for 3D laser fabrication. Finally, we should consider the possibility to mix photoresists with nanoparticles endowed with photothermal, piezoelectric, or magnetic responses. There are, therefore, plenty of possibilities to develop biomimetic responsive microstructures. The literature is vast: we will provide only recent examples on 3D photo-polymerized hydrogels, leaving more exhaustive analysis to previous reviews [133,134,135].

#### 3.7.1. pH-Dependent Micro-Hydrogels

A representative example of a pH-sensitive hydrogel used in TPL writing is BSA-based hydrogel due to its high water solubility and availability. At the isoelectric point (pH ≅5) , BSA is hydrophobic, prone to aggregation, and sticky with respect to glass surfaces. At this pH, water is less extensively coordinated around the protein residues, resulting in the shrinkage of a protein gel, while, at more acidic or basic pH values, the protein chains become ionized: this facilitates hydration and intermolecular repulsion, thereby causing the protein structures to swell.

To give a few examples of the possibility of achieving pH-based mechanical actuation of BSA hydrogels, we consider dynamic and reversible BSA-based deformable microstructures. Y-shaped or in-homogenous structures have been printed [136] by varying the cross-linking density or by exploiting the pH swelling to induce reversible deformations. The authors demonstrated the versatility of the approach by fabricating a Venus fly trap replica [136] in which four tips of a TPP-fabricated structure bent inward to generate a closed trap when pH was raised from 5 to 11. An Erbium-doped, femtosecond laser source (780 nm, pulse width = 100–200 fs) was used with an average power P= 16 mW. The writing parameters were: 0.2 μm line distance and 30 μm/s scan speed. Only the layer distance was varied to change the rigidity of the hydrogel matrix [136]. By patterning microstructures with low and high cross-linked density and by exploiting the pH-induced swelling, Lay et al. investigated a series of possible dynamic morphological changes in BSA hydrogel microstructures’ transitions between circle and polygon [137]. More complex conformational changes were also demonstrated in other BSA-based 3D microstructures [138].

Other hydrogels have been devised for pH-sensitive micro-fabricated structures. PEG polymers, which already have a number of advantages, such as low protein absorption, negligible inflammatory profile, and easy functional modification, can be endowed with pH sensitivity through their coupling with smaller, pH-sensitive molecules. Scarpa et al. [139] used 2-carboxyethyl acrylate (CEA) to confer high-molecular-weight PEGDA (10 kDa) with pH sensitivity and fabricated microstructures that, depending on the geometry, showed either deformations or only a volume change when varying the pH. The authors used Irgacure-2959 as PI and a femtosecond laser for TPL writing at average power of 17 mW, highly focalized in the photoresists (100× magnification objective), and raster-scanned at a speed of approximately 0.1 mm/s.

BSA was also mixed with other organic polymers to endow polymers structurally more rigid than proteins with pH responsivity. For example, BSA proteins were incorporated into poly(methyl methacrylate) (PMMA) for the fabrication of microscopic 3D structures via TPL writing [131]. BSA apparently undergoes a larger swelling ratio change upon pH changes with respect to other proteins, such as lysozyme or avidin [131]. The authors experimented with the protein concentration gradient in the PMMA hydrogels to endow them with expansion and contraction capability, while the protein maintained its capability to participate in electrostatic and hydrophobic interactions. The impressive potential of this technology was demonstrated by devising a pH-activable trap for Esche*richia coli* (*E. coli*) bacteria (Figure 10E): the E. coli could be captured at pH = 7, perfectly alive and capable of growing, and forced to exit when the pH was raised to 12.2 and the inner volume of the chamber decreased.

#### 3.7.2. Photothermally Responsive Microstructured Hydrogels

Light is a particularly suitable agent for remote activation. An example of this possibility is offered by the photothermal effect that consists in the efficient conversion of light into thermal energy with minimal loss as scattering of fluorescence/phosphorescence. Any absorbing molecule has some degree of photothermal effect. Among the most efficient photothermal agents are, however, listed metal and semiconductor nanoparticles that absorb either by inter-band transitions or by plasmonic resonances [140,141]. Among the first studies of the photothermal effect in hydrogels, Palermo et al. [142] demonstrated the possibility to photo-synthesize in situ spherical gold nanoparticles in PVA gels that, when irradiated at 532 nm, showed a photothermal effect ranging from 6.8 to 45 °C/W. Our group more recently fabricated microstructured proteinaceous hydrogels embedding gold nanostars and having photothermal efficiency of up to 10°C/W under continuous wave irradiation at 800 nm [30]. It is worth mentioning that an essential step to obtain photothermally active photoresists is to avoid any chemical interaction or energy transfer between the nanoparticles and the PI in the photoresist. For this purpose, it was essential to adopt Rose Bengal as the PI in TPL writing at 800 nm (average power 100 mW, 10 $\frac{\mu m}{s}$ writing speed) for obtaining photothermal microstructures. The resulting 1.5–2 °C temperature increase, likely underestimated by the average due to the large point spread function width of the Germanium optics of the thermal camera, was measured at high spatial resolution, ≅50 μm, by means of photo-activated subdiffraction thermal imaging [143].

The photothermal effect has been used for cancer therapy for decades and implemented also through metal nanoparticles [140], and photothermal hydrogels can be advantageously used in this area of research, as outlined by the study of Luo et al. [95], who developed bifunctional scaffolds based on a mixture of dopamine-modified alginate and polydopamine that was fabricated using a 3D printer with layer-by-layer deposition and developed by soaking in a manganese chloride solution. Though the scaffolds were not obtained by laser printing, this is an example of a future application of these classes of hydrogels for medical therapy. A second and similar example was presented more recently [144] by Zhang et al., who used a novel class of poly(N-isopropyl acrylamide)/graphene oxide nanocomposite hydrogel mixed with the PI Duracure 2959. Three-dimensional printing was obtained with a commercial 3D filament printer modified with an air pressure controller to extrude the ink from a needle, and the cross-linking was obtained by irradiating at 405 nm (spatial resolution of approximately 120 micrometers). The maximum photothermal effect that could be reached was approximately 23 °C/W for 1 mg/mL graphene oxide concentration. These studies make the field of photothermally activated hydrogels very promising for applications in nanomedicine and call for future efforts in the fabrication of photoresists that could be used for the microfabrication of photothermal, microstructured hydrogels.
sensors-21-05891-t003_Table 3Table 3Summary of the applications of multiphoton lithography to hydrogels treated in the review.MaterialTechniqueSourceApplicationRef.ACRL-PEG-peptide or low-MW PEGDA, 300 mg/mL 2,2-dimethoxy-2-phenyl-acetophenone in N-vinyl-pyrrolidone (NVP)Mask photo-lythography (single photon) and MPLFor MPL, Ti:sapphire, λ= 720 nm, average I = 225 mW/
μm2 pixel dwell time = 120 μ
s/μm2
Softening the matrix; used to guide cell migration[84]PEGDA (575 kDa)9%, 1 nM AuNRs, 0.05% (*w*/*v*) LAPMPL in prefabricated hydrogel by thermal erosionλ=800 nm, fR=80 MHz, ω0=2 μm, P = 80 mW, vS = 1.0 mm/s Modulation of stiffness [87] Irgacure 819, (1 wt%), 20 wt % allyl-functional PNIPAm, 20 wt % NIPAm (Iso-propyl-acrylamide), and 2wt % *N*,*N*′-methylene bis-acrylamide MPL λ=780 nm, τP=120 fs, fR=100 MHz, ω0≅1 μm, P = 30–100 mW, vS = 1000 to 10,000 μm/s3D soft actuator composed of thermoresponsive polymers[88]PEGdiPDA linked to a monoacrylated PEG macromer through a photolabile ortho-nitrobenzylether (NBE)MPL fabrication and erosionTi:Sapphire, λ= 740 nm, pixel dwell time = 1.58 ms, P = 0.1 W Photodegradation of hydrogel to guide hMSC migration[67]Four-arm PEG tetracyclooctyneThermal gelation + MPL erosionTi:Sapphire, λ=
860 nm, P=350–670 mW/μm2, scan speed=1.27 s/μm2)channels of varying depths (150–600 μm); outgrowth of 3T3 fibroblast cells[3]Collagen/fibrin functionalized with a photocaged alkoxyamine. Photocaging through a 2-(2-nitrophenyl) propoxycarbonyl group.Fabrication by photomask UV lithography + MPL erosion Photomask patterning; λ=365 nm, 10 mW/cm2, 10 min exp. time; MPL with lympus FV1000 MPE BX61 Microscope (λ = 740 nm, 0 to 100 laser power, a.u.)Decoration of cell-laden natural hydrogels with growth factors[96]Norbornene-functionalized PVA 22 kDa, 7% substitution; 5–20% in mass, dithiothreitol (DTT) Gel with ≈1 min UV irradiation (365 nm, 10 mW/cm2) and MPL degradationFemtosecond laser (80 MHz, source power: ca. 120 mW at 780 nm), laser power 9 mW, writing speed from 100 to 2000 μm/sPhotochemical construction and manipulation of 3D cellularmicroenvironments of kidney epithelial cells[100]Acrylated photolabile moiety to PEG-bis-amine to create a photocleavable cross-linking diacrylate macromer; TEMED with ammonium persulfateSpontaneous ammonium sulfate polymerization and MPL erosionTwo-photon 740nm laser (3W laser, 20× objective NA~0.75, 1 μm scan intervals over~ 150 μm thickness, laser power = 50%, scan speed setting = 8)Migration of fibrosarcoma cells along the edge of a channel[83]Barnase complexed agarose hydrogel modified with coumarin-caged thiolsUV (365 nm) photolithography followed by MPL photodegradation/activationMultiphoton laser (Mai-Tai, Newport) set to λ= 740 nm3D patterns of proteins and growth factors cultures, to guide cells[104]75% v/v of native bovine dermis type-I collagen solution and 25% *v*/*v* MatrigelThermal polymerization followed by MPL erosionτP=1-ns pulses, fR = 100-Hz, at 355 nm), 10 ×/ 0.25 NA objectiveInduction of intestinal stem cells to form tube-shaped epithelia[35]AgaroseCold gelation and thermal laser-induced erosion of agarose hydrogels1064 nm Nd:YAG laser, 100 mW, ω0≅1 μmRat hippocampal cells forming fiber connections between chambers[36]AgaroseCold gelation and thermal laser induced erosion of agarose hydrogelsAn excimer laser (193 nm, ArF) at a spot size of 30 μm and EP ≅ 1 μJ, fR=10 HzArtificial growth of neural tissue[72](3-trimethoxy-silylpropyl) diethylene triamine in a uniform layer of octadecylsilane Laser ablation of self-assembled monolayers in presence of laminin-1λ= 800 nm, τP=120 fs, fR=1 kHz, P = 0.5 mWGuide the growth of neurites up to 34 μm length[109]Hybrid poly(N-isopropylacrylamide) (PNIPAm)/cellulose nanofibrils (CNFs) hydrogel, (TEGDMA) as a cross-linkerUV photolithography0.5 J/cm2 UV radiation (365 nm)Hydrogel-based micro-optics[114]BSA and methylene blueMPL fabricationfR= 80 MHz, τP=120 fs, λ= 800 nm, I=60 mW/μm2Hydrogel-based micro-optics, kinoform lenses[52]BSA and methylene blueMPL fabricationfR= 80 MHz, τP=120 fs, λ= 780 nm, I=30–60 mW/μm2Microlenses with pH-dependent focal length [115]BSA and methylene blueMPL fabricationfR= 80 MHz, τP=120 fs, λ= 780 nm, P=15 mW, dwell time 1000 μs ω0 ≅ 1 μmTunable harmonic diffractive lenses (HDL)[116]PEGylated fibrinogen hydrogelsMultiphoton LA in hydrogelsEither τP=1 ns, fR=100 Hz, λ= 355 nm, vS=0.097 mm/s, I=0.5–50 μW/μm2 or τP=100 fs, fR=80 MHz, λ= 880 nm, vS=0.1 mm/s, I=48–544 mW/μm2Design of nerve guidance conduits [122]PEGDA hydrogelMultiphoton LA in hydrogelsτP =140 fs, fR= 80 MHz, λ= 790 nm, fluence ≅37 nJ/μm2Vascular bed created by degradation of the gel and seeding of endothelial cells (ECs)[69]Collagen (7.5 mg/mL), fibrin (10 mg/mL), and a hydrogel containing 2% (*w*/*v*) each of agarose and gelatinMultiphoton LA in hydrogelsτP=140 fs, fR= 80 MHz, λ= 790 nm, P=100–244 mW, dwell time=2 μs; ω0≅1 μm.Connections among channels[123]Collagen hydrogels Multiphoton LA in hydrogelsτP≅ 150 fs, fR= 80 MHz, λ= 830 nm, P=400 mW, dwell time=2 μs; ω0≅ 1 μmStudy of metastasis infiltration[71]PEG hydrogels: acrylate-PEG-(GGGLGPAGGK-PEG)n-acrylateMultiphoton LA in hydrogelsλ= 720 nm, I=203 mW/μm2, vS=240 μs/μm2 scanning rateControlled HDF migration[124]PEGDA hydrogel (with IrgacureTM PIs) or BSA with RiboflavinMultiphoton LA in hydrogelsτP≅ 94 fs, λ= 780 nm, fR=80 MHz P=0.9–2.64 mW ω0≅1 μm, vS=10–30 μm/s*E. coli* trapping[125,126]PEGda, 700Da) 50 wt%, 2 wt% PI: 2PCK or DASMPL fabricationτP≅ 150 fs, λ= 800 nm, fR= 80 MHz P=60–410 mW, vS=1–100 mm/sTrapping of cells: osteosarcoma and endothelial cells [128]Methacrylamide (RCPhC1-MA), norbornene (RCPhC1-NB), or thiol (RCPhC1-SH) + DASMPL fabricationτP≅ 70 fs, λ= 720 nm, fR= 80 MHz P=10–100 mW, vS=1000 mm/sEncapsulation of human adipose stem cells[129]PEGDa and a water-soluble PIMPL fabricationτP≅ 100 fs, λ= 810 nm, fR= 73 MHz, P=400 mW; vS=10 mm/sEncapsulation of *Caenorhabditis elegans (C. elegans)*[132]PEG–coumarin hydrogelsMPL fabricationτP≅ 100 fs, λ= 700–850 nm, fR=80 MHz, P =0.8 mW; vS=0.8 mm/sIn vivo subcutaneous fabrication of hydrogels[130]BSA hydrogelsMPL fabricationerbium doped, fs laser (λ= 780 nm, τP=100–200 fs), P=16 mW, vS=30 μm/s.pH-sensitive deformable structures[136,137]2-carboxyethyl acrylate (CEA) and PEGDA (10 kDa), PI = Irgacure-2959MPL fabricationTi:Sapphire, λ= 780 nm, P = 17 mW, vS=100 μm/s.pH-sensitive deformable structures[139]BSA and Rose Bengal and gold nanostarsMPL fabricationτP≅ 250 fs, λ= 780–800 nm, fR=80 MHz, P=100–200 mW; vS=010 μm/sIn vivo subcutaneous fabrication of hydrogels[130]

## 4. Conclusions. Two in One—Light Brush and Light Chisel for Hydrogels

Using a pen to write and an eraser to correct the text are common actions in daily life. A fountain pen exploits the balance between capillary and adhesion forces, tuned by the hydrophilicity of the ink. This old technique is the basis of some nano-writing approaches, such as dip pen manolithography [145]. The eraser mechanically scrapes material from the cellulose: similarly, techniques such as focus ion beam lithography can ablate the atomic layers of a solid surface. In the chemistry of materials, the writing and erasing concept can be translated into the formation and cleavage of bonds, respectively. These phenomena are very different from the original pen/eraser action, since the writing is due to polymerization or cross-linking and erasing to ablation that occurs when the material absorbs sufficient energy to melt or vaporize. The first additive step can be well regulated, while the second is a chemically undefined process [65,146]. In this review, we wished to show that laser microstructuring is an extremely versatile tool for the development of microstructured hydrogels, particularly if polymerization (writing) and ablation (erasing) can be combined in the same platform [147]. In addition, by also exploiting chemical etching by aggressive or plain (water soluble) solvents [148], we can gain even more flexibility, particularly in terms of the variety of accessible space scales.

Among subtractive fabrication methods, fs laser ablation (LA) has been applied to a wide variety of transparent materials [149]. When LA is applied to transparent materials, structural or phase modification of the materials occurs, leaving behind permanent changes in the refractive index or even voids. Laser ablation by means of femtosecond lasers is advantageous for many reasons, mainly due to the nonlinear interaction that is exploited and the consequent reduction in the diffused thermal effect, which allows highly localized and 3D controlled microstructuring to be achieved. Much lower pulse energies are needed for hydrogels (approximately micro-Joules) [5] with respect to glasses [150] and polymers [149]. Interestingly, LA was used also in nanosurgery to ablate tissue with high precision and, recently, in opto-genetics [151,152]. A thorough discussion of the literature production has been reported in a few recent reviews [65,149]. As we have reviewed here, only a few examples of coupled MPL writing and LA have been presented in the past 10 years. Xiong et al. [153] showed the possibility to employ the same laser radiation to both prime photo-polymerization and induce ablation in a highly nonlinear absorption regime. In their work, Xiong et al. integrated additive (MPL) and subtractive (LA) processes using a commercial fs laser direct writing system (Nanoscribe GmbH). By employing an 80 MHz, 120 fs, Ti:Sapphire laser system with a proprietary resin (IP-L, Nanoscribe), they were able to produce fibers (polymerization) and holes (ablation) with width/diameter down to 200 nm. They claimed that such high resolution was due to the three-photon regime for the IP-L resin.

Interestingly, filament extrusion fabrication (FFF) has also been combined for the creation of microstructures having a multiscale spatial distribution with an internal 3D microstructure geometry [154]. The manufactured microporous structures formed from polylactic acid (PLA) had fully controllable porosity (20–60%, porous size of around 0.5 μm). The prepared scaffolds were biocompatible with various cell lines, and primary stem cell growth could be observed on the printed structures. It is interesting that the surface of the structures can be made more hydrophilic for cell growth simply by dipping the microstructures in acetone, a procedure that enhances the porosity of the structures.

In addition, LA was employed to modify PLA structures—for example, by drilling holes by means of filamentous propagation in water. The combination of FFF printing with LA, proposed by Malinauskas and coworkers [154], paves the way towards the laser production of meso-structures whose micro-architecture is important for cell culturing. The group used an average optical power P = 5–100 mW and pulse energy Ep = 0.2–4 μJ (corresponding to I = 4.2–83 $\mathrm{TW}/{\mathrm{cm}}^{2}$) for LA processing. Such high energies cannot be reached with a Ti:Sapphire laser, which is the most widely used for hydrogel MPL. However, for proteinaceous hydrogels, lower pulse energy values are typically required for LA, and combining different approaches, such as FFF, MPL, and LA, would be a winning strategy. It would be highly desirable in the next few years to be obtain high-resolution, single-cell sensing in longitudinal studies similarly to the current sophisticated millifluidic [155] and microfluidic setups [94]. This, if achieved, would be beneficial to a large number of areas in nanomedicine and nano-biotechnology, such as tissue engineering from lab research to the clinical bed. The possibility that 3D-printed, hydrogel-based devices will be, in the future, a standard for personalized prostheses and drug dispensing will also have a wide economical impact. The use of additive manufacturing with respect to conventional manufacturing in bedsized or point-of-care medical devices can be widened and diversified. The production could be distributed to retailers (possibly spin-offs of research institutions and clinics) if the cost of the 3D printer does not impact too much on the cost of each printed device. In this way, direct and rapid access to personalized devices would also be possible, with a social impact that could balance the economical burden. Many aspects of these complex scenarios have been explored by Arbabian and Wagner [156], though only for the automotive and opto-electronics fields. No similar study is still present in the literature for the case of medical therapy and diagnosis.

Returning to the R&D impact of 3D manufacturing in health science, even though many possibilities are offered by the coupled use of laser photo-polymerization and photo-ablation, and by the combination of mesoscopic FFF with high-resolution 3D MPL printing, we can single out a few caveats that should addressed in the future: (1) 3D additive/subtractive manufacturing can be extremely slow; (2) there are too many parameters affecting the hydrogels’ features; (3) printing devices based on MPL can be too expensive to be used widely in many laboratories. Indeed, we have too many parameters to explore to obtain hydrogels with the desired porosity, stiffness, chemical reactivity etc. In this regard, it would be promising to use machine learning approaches, which could overcome the lengthy sequential exploration of the whole parameters’ phase space and speed up significantly the fabrication of the desired microstructures. This approach has been recently discussed by Nasiri and Khosravani [157]. Regarding the technological costs, instead, additional efforts should be made in the future to reduce not only the source costs, but also those of reliable 3D scanners essential for accurate 3D printing.

## Figures and Tables

**Figure 1 sensors-21-05891-f001:**
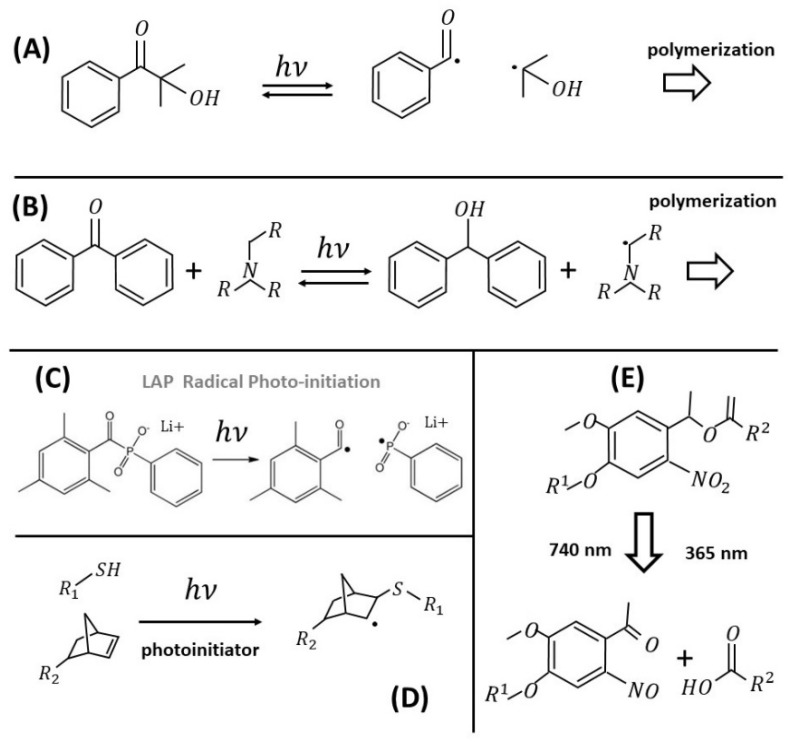
Norrish type of photoinitiation of the radical polymerization for type I photoinitiators (**A**) and type II photoinitiators (**B**). (**C**) represents schematically the photoinitiation of the PI LAP (see Table 1). (**D**) represents schematically the photoinitiation of one common photo-click reaction based on the norbornene moiety. (**E**) represents one common photo-cleavage reaction involving the NBE group (see Table 1).

**Figure 2 sensors-21-05891-f002:**
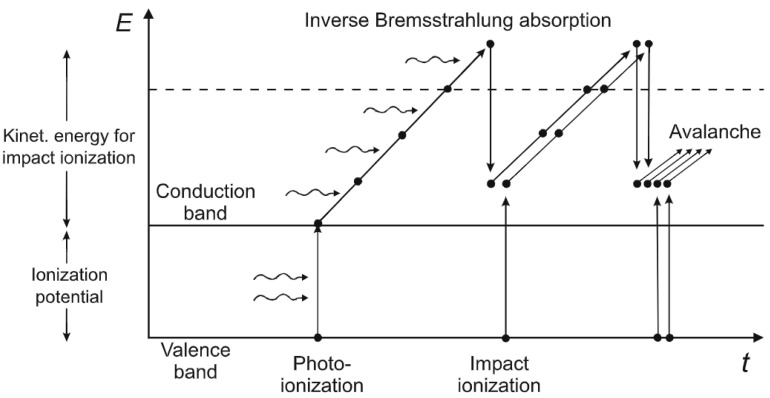
Schematics of the general mechanisms at the origin of the energy deposition in polymeric materials. Excitation into the conduction band (ionization) leads to quasi-free electrons still loosely bound to atoms. The excitation energy into the conduction band can be provided either by MPI or TI, depending on the Keldysh parameter, or by impact ionization from an energetic quasi-free electron that has gained its energy through photon absorption in a non-resonant process called ‘inverse Bremsstrahlung’ in the course of collisions with ions. This process is also at the origin of the avalanche ionization. Reprinted with permission from [78].

**Figure 3 sensors-21-05891-f003:**
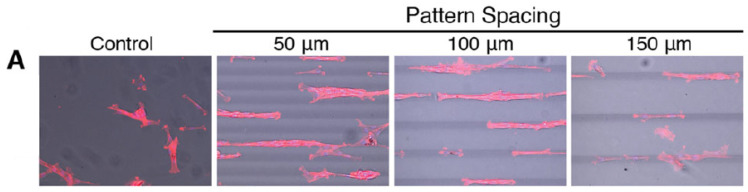
Stiffness-modulated hydrogels with 50, 100, 150 micron spacing between the patterns. A7R5 smooth muscle cells show migration to the stiffer regions patterned by MPL patterning. A7R5 cells stained for actin/nuclei on day 6 after incubation (adapted and printed with permission from [87]).

**Figure 4 sensors-21-05891-f004:**
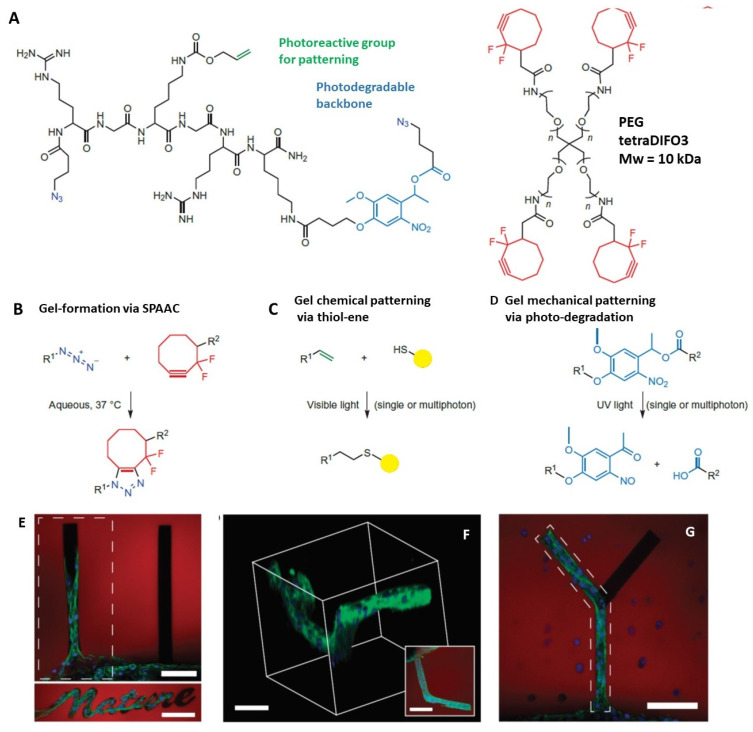
Photocoupling and photodegradation based on click chemistry in hydrogels. A, B, Click-functionalized macromolecular precursors (PEG-tetraDIFO3 and bis(azide)-functionalized polypeptides) can form a 3D hydrogel structure (**A**) by means of a step-growth polymerization mechanism via (**B**) the SPAAC reaction. In the presence of visible light (λ≅ 490–650 nm or 860 nm, single or two-photon excitation), thiol-containing biomolecules are covalently linked (**C**) to pendant vinyl functionalities throughout the hydrogel network via the thiol–ene reaction. A nitrobenzyl ether moiety within the backbone of the polymer network undergoes photocleavage (**D**) upon single- or multiphoton absorption (λ≅ 365 nm or 740 nm), resulting in photodegradation of the hydrogel. (**E**): a fibrin clot containing 3T3 fibroblasts was encapsulated within the click hydrogel formulation. Chemical channels coated with RGD, a cell-adhesive fibronectin motif, were created radially out of the roughly spherical clot (bottom of the image). By day 10, cells were found to migrate only down the physical channel functionalized with RGD. (**F**): by creating 3D functionalized channels, cell outgrowth was controlled in all three spatial dimensions, with the image inset illustrating a top-down projection. (**G**): the outgrowth of 3T3 fibroblast cells was confined to branched photodegraded channels functionalized with RGD. The regions of RGD functionalization are highlighted by the dashed polygons in E and F. Hydrogel is shown in red, F-actin in green, and cell nuclei in blue. Scale bars, 100 μm. Adapted and reprinted with permission from [3].

**Figure 5 sensors-21-05891-f005:**
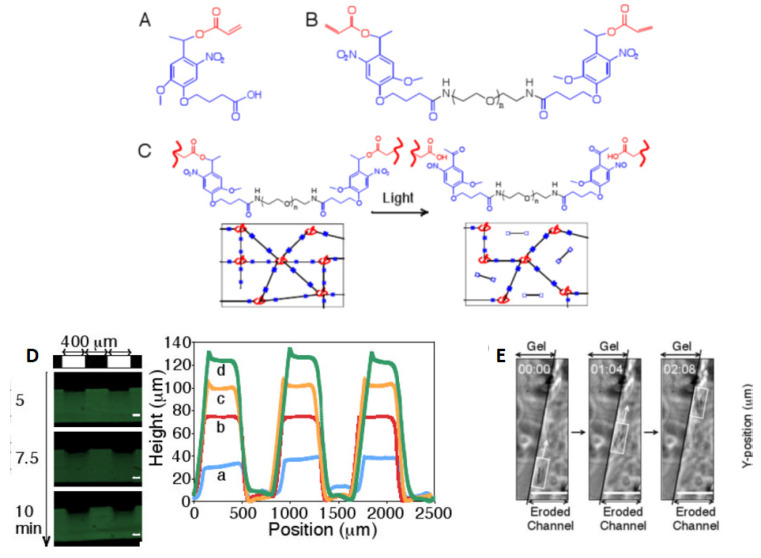
Panels A–C: (**A**) base photodegradable acrylic monomer used to synthesize (**B**) the photodegradable cross-linking macromer (Compound B, Mn∼4070 g/mol), comprising PEG (black), photolabile moieties (blue), and acrylic end groups (red). (**C**) Compound B was copolymerized with PEGA (Mn∼375 g/mol), creating gels composed of poly(acrylate) chains (red coils) connected by PEG (black lines) with photolabile groups (solid blue boxes). (**D**): Thick gel surface erosion induced by irradiation (left; scale bar, 100 μm). The gel, covalently labeled with fluorescein, was eroded spatially via masked flood irradiation (320–500 nm at 40 mW/${\mathrm{cm}}^{2}$). Channel depth, quantified with profilometry, increased linearly with irradiation time: (a) 2.5, (b) 5, (c) 7.5, and (d) 10 min (right). (**E**): Channels were eroded within a hydrogel encapsulating fibrosarcoma cells, releasing cells into the degraded channel and enabling migration. Migration of a cell along the edge of a channel is shown in time-lapsed brightfield images. Scale bar, 50 μm. Adapted and reprinted with permission from [83].

**Figure 6 sensors-21-05891-f006:**
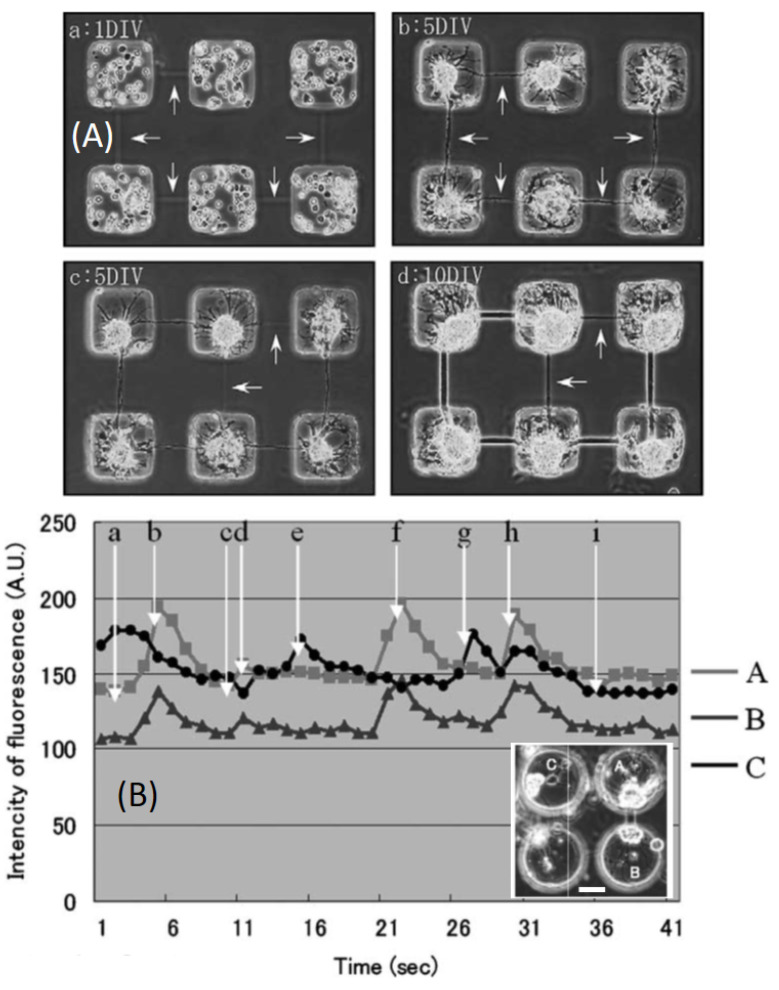
(**A**): example of the effect of photothermal etching during hippocampal cell cultivation. (a) and (b) compare situations immediately after photothermal etching and 5 days after the treatment, when (c) a new tunnel (white arrow) was photothermally created between agarose microchambers, leading, after another 5 days (d), to new neurites connecting cells through the newly added tunnel. (**B**): time dependence of Ca fluorescent signals from neural cells. Lines A, B, and C indicate the signals of the corresponding cells/microchambers in the image inset. Adapted and reprinted with permission from [36].

**Figure 7 sensors-21-05891-f007:**
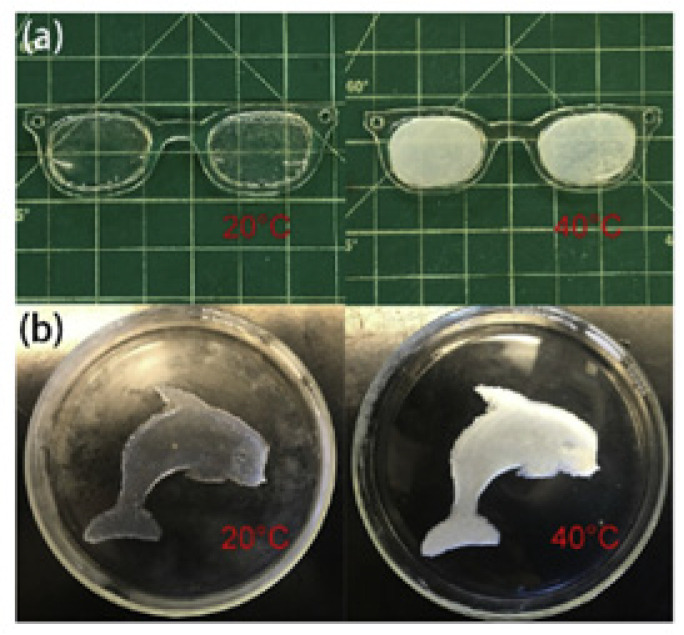
3D-printed PNIPAm/CNF hydrogels at T = 20°C  (**left**) or T = 40 °C (**right**) in the shape of lenses (panel **a**) or a dolphin (panel **b**). Reprinted from [114], with permission.

**Figure 8 sensors-21-05891-f008:**
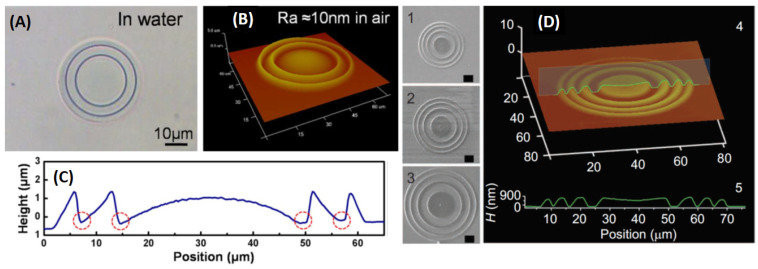
(**A**): Optical microscopy picture of a protein harmonic diffractive microlens in water; (**B**): atomic force microscopy (AFM) image of the microlens in (A). Roughness average is ∼10 nm; (**C**): section contour of the microlens in (A) characterized by AFM imaging in air. (**D**): SEM images of protein micro-KPLs on a glass coverslip with 1 μm thickness but different diameters, (1) 50 μm, (2) 60 μm, and (3) 80 μm. Scale bar 10 μm. (4) AFM characterization exhibiting the 3D morphology and the 10 nm average roughness of the protein micro-KPL (diameter 60 μm). (5) The thickness of the protein micro-KPL at the central line in (4) as measured by AFM. Panels A-C, adapted and reprinted with permission from [116]. Panel D adapted and reprinted with permission from [52].

**Figure 9 sensors-21-05891-f009:**
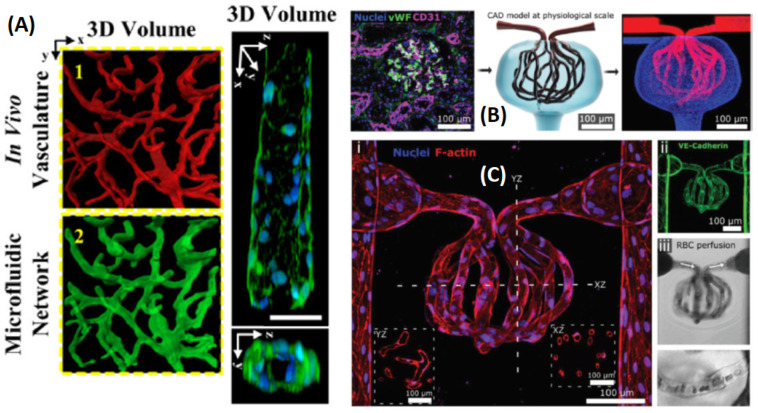
(**A)**, Left: a microfluidic network is fabricated following the in vivo visualization of a mouse cerebral vessel bed to demonstrate the ability to recapitulate the dense, tortuous in vivo vascular network in PEGDA. Right: microchannels seeded with mouse brain endothelial cells, fluorescently labeled with DAPI (blue: nucleus) and ZO-1 (zona occludens protein-1; green: tight junctions), and imaged via confocal microscopy. Bar = 50 μm. (**B**): A human glomerulus (left) informed creation of a CAD photomask (middle) for collagen ablation. Image (right) shows ablated structure following bead perfusion of capillaries (red) and Bowman’s space (blue). (**C**): cellularized glomerular structure. (i) 3D projection of glomerulus with inset cross-sectional views through the YZ and XZ plane, respectively; (ii) VE-cadherin staining; (iii) RBC perfusion, with magnified view showing single-cell transit (bottom). Panel A adapted and reprinted with permission from [69]; (**B**,**C**) reprinted from [123].

**Figure 10 sensors-21-05891-f010:**
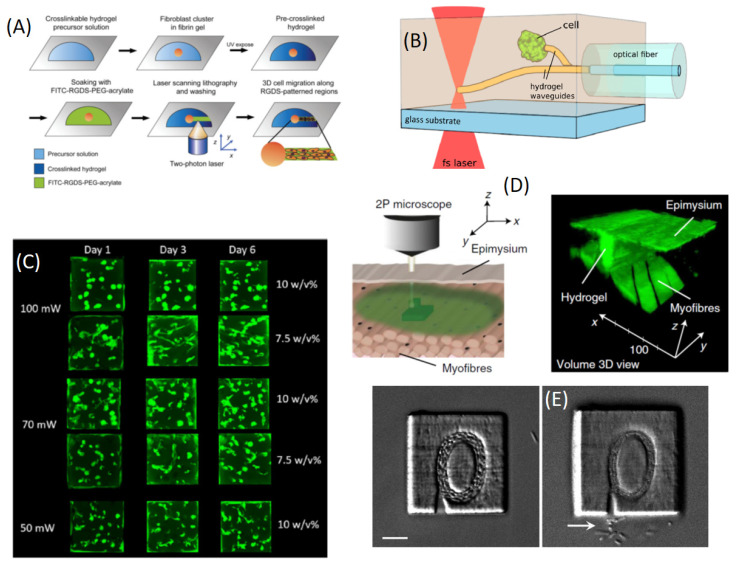
(**A**): The overall methodology for three-dimensional RGDS patterning by TPL: (1) HDFs encapsulated in fibrin clusters are photo-polymerized into collagenase-sensitive PEG hydrogels (385 nm light), (2) hydrogels are soaked in PEG–RGDS solution; (3) TPL is used to pattern RGD rich regions in the hydrogels; (4) after washing, cell migration is monitored over time. Reprinted from [124]. (**B**): Schematic illustration showing the cross-linking process in a cell-loaded hydrogel with a waveguide starting from a fiber connecting a cell for fluorescent excitation. Reprinted from [125]. (**C**): LSM images of 200 × 200 × 200 μm^3^ RCPhC1-NB/SH-based cubes printed in the presence of living ASC–GFPs using 2 mol % DAS with respect to the photo-cross-linkable functionalities (left). The cubes were printed using different concentrations and laser powers. Reprinted with permission from [129]. (**D**): Schematic showing two-photon cross-linking of HCC–hydrogel into skeletal muscle across epimysium (left). Right panel: representative 3D volume reconstruction of 6 independent replicates showing HCC–8-arm PEG structure (Δ*z* = 300 μm) manufactured between undamaged myofibers and epimysium of skeletal muscle in GFP+ mice; coordinates and 50 μm scale bar are shown. From [130]. Panel (**E**): Differential interference contrast image of E. coli cells densely packed into an o-shaped chamber after several cell divisions. Abrupt change in bath pH (7 to 12.2) causes temporary compression of the chamber and the release of a few cells (arrow). (Scale bars, 5 μm). Reprinted with permission from [131].

**Table 1 sensors-21-05891-t001:** Acronyms, symbols.

AFM	Atomic force microscopy	AuNR	Gold nanorods
BSA	Bovine serum albumin	CEA	2-carboxyethyl acrylate
CNF	Cellulose nanofibril	CuAAC	Copper(I)-catalyzed azide-alkyne cycloaddition
DAS	Tetrapotassium 4,4′-(1,2-ethenediyl) bis[2-(3-sulfophenyl) diazenesulfonate]	ECM	Extracellular matrix
FFF	Filament extrusion fabrication	HA	Hyaluronic acid
HCC	7-hydroxycoumarin-3-carboxylic acid	HDF	Human dermo-fibroblasts
HDL	Harmonic diffractive lenses	ITX	2-/4-isopropylthioxanthone
LAP	Lithium phenyl-2,4,6-trimethyl benzoylphosphinate	LA	Laser ablation
LSL	Laser-scanning lithography	MAP	Multiphoton absorption polymerization
MPI	Multiphoton ionization	MPL	Multi-photon lithography
MSC	Mesenchymal stem cells	NIR	Near-infrared
NBE	Ortho-nitrobenzylether	PAG	Photoacid generator
PDMS	Polydimethylsiloxane	PEG	Poly-ethylene glycol
PEGDA	PEG diacrylate	PET	Poly(ethylene terephthalate)
PI	Photoinitiator	PLA	Polylactic acid
PMMA	Polymethyl methalcrylate	PS	Photosensitizer
PVA	Poly-vinyl-alchool	PNIPAm	Poly(N-isopropylacrylamide)
TI	tunneling ionization	TPA	Two-photon absorption
TPP	Two-photon polymerization	RGDS	Arginylglycylaspartic acid
SEM	Scanning electron microscopy	SPAAC	Strain-promoted azide-alkyne cycloaddition

**Table 2 sensors-21-05891-t002:** Physical symbols describing the main features of pulsed lasers used for MPL fabrication.

I	Average intensity	w0	Laser beam waist
vscan	Scanning speed	P	Average power
σ2	Two-photon cross-section	fR	Laser repetition rate
τP	Laser pulse width	v3D	Rate of volume fabrication
τV	Voxel time (=w0vS)

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
