# Peer review of "Multiphoton Laser Fabrication of Hybrid Photo-Activable Biomaterials"

_sensors, 2021, doi:10.3390/s21175891_

Round 1

Reviewer 1 Report

Dear Authors,

in your interesting review, the following points should be added/changed to further improve it:

  • Table of contents, tables with abbreviations and physical symbols: I am not sure whether they fit to the journal format ... but this can surely checked afterwards. However, please check the full meanings of SPAAC and DAS. And what does the doubled equal sign in the definition of the voxel time mean? Besides, please use w_0 instead of omega_0 for the beam waist; the omega normally means something completely different.
  • line 60: fineness instead of finesse
  • line 159: polyvinyl pyrrone
  • Fig. 1: Some of the smaller letters are hard to read and seem to be written with different colors. Are the slight rotations of the rings in Fig. 1(e) real or just an artifact due to decreasing the image?
  • line 214: I know the abbreviation Ti:Sa or the full description Ti:sapphire, but this abbreviation here is a little bit unusual - please think about changing it (ditto in line 327, Table 3 and some other positions).
  • line 423: Here you write "(Figure 2) [79]", while the figure caption is related to [80] - please check.
  • Fig. 4a: Is it possible to write the "for" without capital F?
  • Fig. 6: In the caption, the reference is missing; ditto in Fig. 9.
  • Fig. 8: Here again, references in the text and in the figure caption seem to be different.
  • lines 827, 830: "type I collagen gel" is doubled.
  • line 969: The nematode misses in "i".
  • line 1045: Please mention the full name of E. coli firstly and write "coli" without capital C.
  • Table 3: Just a formatting suggestion - actually this table may be broader, so it may make sense to play a little with column widths to make it a little bit shorter.
  • And finally, please check the references. They are not formatted according to the journal template, and a few (e.g. [5]) have some additional errors.

Author Response

We thank the Reviewer for the criticisms and the suggestions that have now been taken into account in the revised form of the Ms. Please, find the answers in the attached pdf file.

Reviewer 2 Report

There are some weaknesses through the manuscript which need improvement. Therefore, the submitted manuscript cannot be accepted for publication in this form, but it has a chance of acceptance after a minor revision. My comments and suggestions are as follows:

1- Abstract gives information on the main feature of the performed review, but some details about the main concepts of previous research works must be added.

2- I think table of content is not required.

3- Authors must clarify necessity of the performed research. Objectives of the study must be clearly mentioned in introduction.

4- Considering 3D printing methods literature study must be enriched. In this respect, authors must read and refer to the following papers: (a) https://doi.org/10.1016/j.jmrt.2021.07.004  (b) https://doi.org/10.1016/j.ejor.2020.01.063

5- The current version of paper is too long. 

6- The main reference of each formula must be cited. Moreover, each parameters in equations must be introduced. Please double check this issue.

7- In its language layer, the manuscript should be considered for English language editing. There are sentences which have to be rewritten.

8- The conclusion must be more than just a summary of the manuscript. List of references must be updated based on the proposed papers. Please provide all changes by red color in the revised version.

Author Response

(The authors gave the same response as above.)
